# Global predictions for the risk of establishment of Pierce's disease of grapevines

Alex Giménez-Romero [1], Javier Galván[1], Marina Montesinos[2], Joan Bauzà [3], Martin Godefroid[4], Alberto Fereres[4], José J. Ramasco [1], Manuel A. Matías [1] & Eduardo Moralejo [2✉]

The vector-borne bacterium *Xylella fastidiosa* is responsible for Pierce's disease (PD), a lethal grapevine disease that originated in the Americas. The international plant trade is expanding the geographic range of this pathogen, posing a new threat to viticulture worldwide. To assess the potential incidence of PD, we have built a dynamic epidemiological model based on the response of 36 grapevine varieties to the pathogen in inoculation assays and on the vectors' distribution when this information is available. Key temperature-driven epidemiological processes, such as PD symptom development and recovery, are mechanistically modelled. Integrating into the model high-resolution spatiotemporal climatic data from 1981 onward and different infectivity ($R_0$) scenarios, we show how the main wine-producing areas thrive mostly in non-risk, transient, or epidemic-risk zones with potentially low growth rates in PD incidence. Epidemic-risk zones with moderate to high growth rates are currently marginal outside the US. However, a global expansion of epidemic-risk zones coupled with small increments in the disease growth rate is projected for 2050. Our study globally downscales the risk of PD establishment while highlighting the importance of considering climate variability, vector distribution, and an invasive criterion as factors to obtain better PD risk maps.

[1] Instituto de Física Interdisciplinar y Sistemas Complejos, (IFISC-UIB-CSIC), Campus UIB, 07122 Palma de Mallorca, Spain. [2] Tragsa, Passatge Cala Figuera 6, 07009 Palma de Mallorca, Spain. [3] Departamento de Geografía, Universidad de las Islas Baleares, Campus UIB, 07122 Palma de Mallorca, Spain. [4] Instituto de Ciencias Agrarias, Consejo Superior de Investigaciones Científicas, ICA-CSIC, 28006 Madrid, Spain. ✉email: emoralejor@gmail.com

Emerging plant pathogens and pests are costly both economically and environmentally for society[1–4]. Among valuable crops recurrently affected by emerging diseases, the grapevine occupies a remarkable place in the history of plant pathology[5–8]. Nowadays, Pierce's disease (PD) is considered a potential major threat to winegrowers worldwide[9]. The annual economic burden in California alone has been estimated at over $100 million[10], and the disease is a well-recognised limiting factor in the cultivation of *Vitis vinifera* in the southeastern US[9]. In Europe, despite strict quarantine measures to protect the wine industry (Directive 2000/29/EC), PD has recently been established for the first time in vineyards on the island of Majorca, Spain[11,12]. This finding, alongside the detection of PD in Taiwan[13], has raised concerns about its possible spread to continental Europe and other wine-producing regions worldwide.

The causal agent of PD[14], the bacterium *Xylella fastidiosa* (Xf)[15], is native to the Americas where it also causes vector-borne diseases on many economically important crops, such as citrus, almond, coffee and olive trees[16,17]. Xf is phylogenetically subdivided into three major monophyletic clades that correspond to the three formally recognised subspecies: *fastidiosa*, *multiplex* and *pauca*, native from Central, North, and South America, respectively[18,19]. Although as a taxonomic unit Xf infects more than 560 plant species[20], it also shows genetic variation among subspecies and sequence types (STs) in both host specificity and host range[21]. Since 2013, diverse STs of the three subspecies have been detected in Europe mainly associated with crop and ornamental plants[22–24]; among these, the clonal lineage of the subsp. *fastidiosa* responsible for PD (hereafter termed Xf$_{PD}$). The same genetic lineage also causes almond leaf scorch disease in California[25] and Majorca (Spain)[26], where it is widespread in almond plantations and vineyards, affecting more than 23 grape varieties[12].

A key trait in the understanding of Xf's invasive potential is its capacity of being transmitted non-specifically by xylem sap-feeding insects belonging to sharpshooter leafhoppers (Hemiptera: Cicadellinae) and spittlebugs (Hemiptera: superfamily Cercopidae)[27,28]— e.g., at least eight species transmit PD in the southeastern US[29]. Such non-specificity would have facilitated Xf$_{PD}$ invasion after being unwittingly brought to Majorca around 1993 with infected almond cuttings from California and its spread thereafter to grapevines through local populations of the meadow spittlebug, *Philaenus spumarius*[26]. Recently, the role of *P. spumarius* in the transmission of PD in Majorca has been demonstrated[12] and its involvement in epidemic outbreaks in California, previously thought marginal[30,31], is being revisited[32,33]. To date, the meadow spittlebug has been confirmed as the major vector in the olive quick decline syndrome, PD and the almond leaf scorch disease outbreaks in Europe[12,26,28,34]; therefore, its geographic distribution should be taken into account when assessing the risk of Xf-related diseases[35].

The tropical origin of Xf subsp. *fastidiosa* already suggests that PD is a thermal-sensitive disease, with the temperature being a range-limiting factor[36,37]. Thus, the accumulated heat units (i.e., growing-degree days) required to complete the process from Xf$_{PD}$ infection to symptom development is critical to predicting the probability of developing PD acute infections[38]. Conversely, the effect of cold-temperature exposures in the recovery of Xf-infected grapevines is a well-established phenomenon[38–40], limiting the geographic range and damage of chronic PD in vineyards in the US[9]. Such "winter curing" has been linked to the average $T_{min}$ of the coldest month, to exposures to extreme cold temperatures for several days, or to the accumulation of chilling hours[41]. The dynamics of chronic infections—i.e., those that persist from one year to the next year—are determined by the net balance between the number of new infections during the growing season and those infected plants recovered in winter.

Because new infections late in the growing season are more likely to recover during winter than early-season infections, the vector's phenology greatly influences the dynamics of chronic infections and PD transmission[30,42–44].

Several works have attempted to predict the potential geographic range of the subsp. *fastidiosa*[45–47] and other Xf subspecies in Europe[48,49] and worldwide[47] using bioclimatic correlative species distribution models (SDMs). However, none of these works has explicitly included information on vectors' distribution or disease dynamics. They hence provide little epidemiological insight into the underlying environmental causes underpinning or limiting a potential invasion. An alternative to overcome these limitations is to develop mechanistic models based on the physiology of the pathogen[50], coupled with epidemiological models that consider the disease dynamics while avoiding the difficulties of including transmission parameters for each of the PD potential vectors.

Risk maps often represent an average snapshot that overlooks interannual climate variability and the effects of climate change as limiting disease factors per se. This leads frequently to risk overestimation[51–54]. Increased availability of computational resources to deal with demanding climate databases now makes it possible to fit dynamic epidemiological models that include climate variability at broad spatiotemporal scales. For example, high-resolution satellite-based climate data have been employed for testing mechanistic models that relate critical physiological processes of coffee rust with climate variables in past outbreak events[55]. Despite these important advances, no attempt of exploring mechanistic SDM has been performed yet for PD.

In this work, we present a temperature-driven dynamic epidemiological model to infer where PD would have become endemic in different wine-growing regions worldwide from 1981 onward if we forced the introduction of Xf-infected plants. We follow an invasive criterion as defined by Jeger & Bragard[56] to include, as far as we can, key plant, pathogen, and vector parameters and their interactions for estimating the risk of establishment, persistence, and subsequent epidemic development. The model assumes a local Xf$_{PD}$ spatial propagation among plants mediated by the presence of potential vectors. Due to the limited knowledge about the vectors of PD in most wine-growing regions of the world[30], we employ a fixed maximal estimate for basic reproductive numbers ($R_0$) in the epidemiological models, except for Europe, where there are precise estimations of climate suitability for the main vector *P. spumarius*[35]. This heuristic approach to obtaining PD risk maps yields results that are consistent with all the relevant data available[45]. It also allows us to quantitatively approximate the current potential growth rate of PD incidence in wine-growing regions under different transmission scenarios, as well as extrapolate the impact of PD by 2050[57]. By estimating a lower global risk of PD, our study casts doubts on the potential impact predicted for other Xf-related diseases transmitted by *P. spumarius*[49], specially in Europe when vector distribution is taken into account.

## Results

**Thermal requirements to develop PD**. We examined the response of a wide spectrum of European grapevine varieties to Xf$_{PD}$ infection in three independent experiments conducted in 2018, 2019, and 2020. Overall, 86.1% ($n = 764$) of 886 inoculated plants, comprising 36 varieties and 57 unique scion/rootstock combinations, developed PD symptoms 16 weeks after inoculation. European *V. vinifera* varieties exhibited significant differences in their susceptibility to Xf$_{PD}$ (Supplementary Table S1). All varieties, however, showed PD symptoms to some extent, confirming previous field observations of general susceptibility to

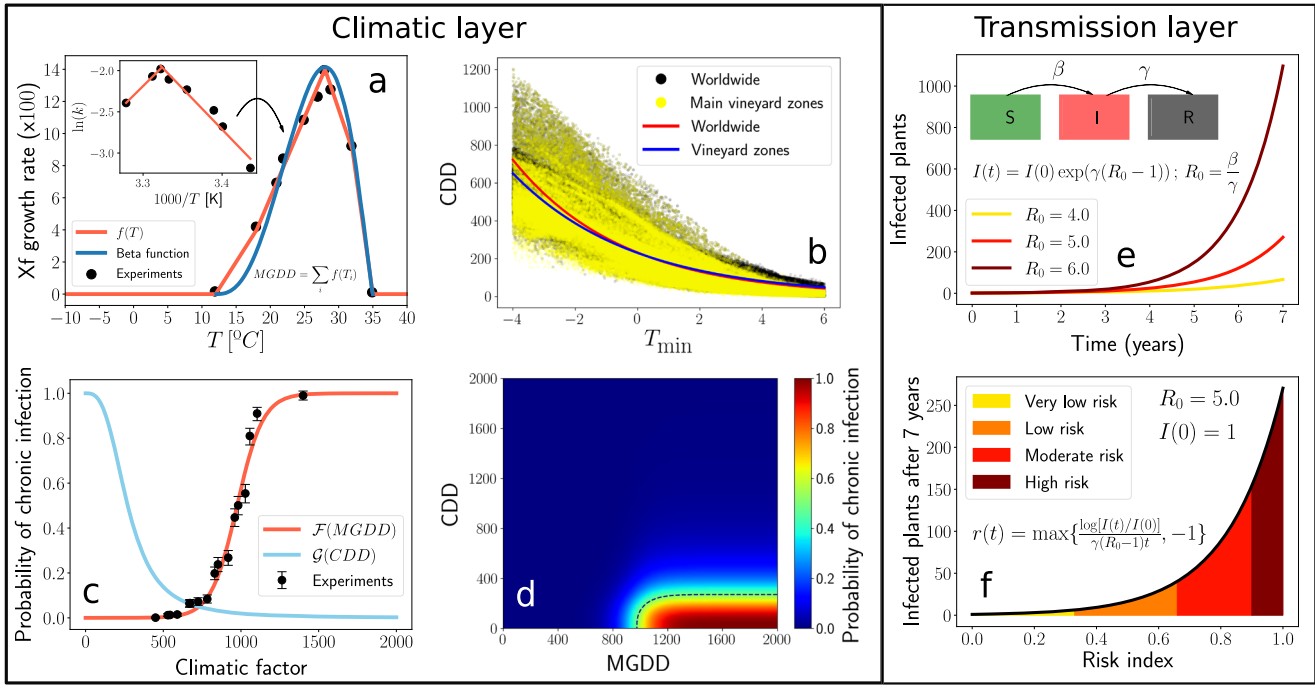

**Fig. 1 Climatic and transmission layers composing the epidemiological model. a** MGDD profile fitted to the in vitro data of Xf growth rate in Feil & Purcell 2001[38]. The original Arrhenius plot in Kelvin degrees (inset) was converted to Celsius, as explained in (Supplementary Note 2A), to obtain the fit shown in the main plot red line; the blue line represents the fit with a Beta function. **b** Correlation between CDD and the average $T_{min}$ of the coldest month between 1981 and 2019. Plotted black dots (worldwide) and yellow dots (main wine-producing zones) depict climatic data from 6,487,200 cells at 0.1° × 0.1° resolution, spread globally and retrieved from ERA5-Land dataset. The red solid line depicts the fitted exponential function for worldwide data and the blue solid line for main vineyard zones. **c** Nonlinear relationship between MGDD (red line) and CDD (blue line) and the likelihood of developing chronic infections. Black dots depict the cumulative proportion of grapevine plants in the population of 36 inoculated varieties showing five or more symptomatic leaves at each of the 15 MGDD levels (see Supplementary Information). Vertical bars are the 95% CI. **d** Combined ranges of MGDD and CDD on the likelihood of developing chronic infection. **e** Transmission layer in the dynamic equation (1) of the SIR compartmental model. **f** Relationship between the exponential growth of the number of infected plants with the risk index and their ranks.

$Xf_{PD}$[9,12,37]. We also found significant differences in virulence ($\chi^2 = 68.73$, df $= 1$, $P = 2.2 \times 10^{-16}$) between two $Xf_{PD}$ strains isolated from grapevines in Majorca across grapevine varieties (Supplementary Fig. S1). Full details on the results of the inoculation tests are available in "Methods", Supplementary Note 1, Supplementary Table S1 and Supplementary Data 1.

Growing degree days (GDD) have traditionally been used to describe and predict phenological events of plants and insect pests, but rarely in plant diseases[58]. We took advantage of data collated in the inoculation trials together with temperature to relate symptom development to the accumulated heat units at weeks eight, 10, 12, 14, and 16 after inoculation (Supplementary Data 1). Rather than counting GDDs linearly above a threshold temperature, we consider Xf's specific growth rate in vitro within its cardinal temperatures. The empirical growth rates come from the seminal work by Feil & Purcell[38] shown in the inset of Fig. 1a. This Arrhenius plot was transformed, as explained in Supplementary Note 2A, to obtain a a piece-wise function $f(T)$ Eq. (1). Our model and risk maps are based on $f(T)$ (red line in Fig. 1a) because it provides the best fit to the experimental data when compared with the commonly used Beta function (blue line) for representing the thermal response in biological processes[59,60]. This Modified Growing Degree Day (MGDD) profile Eq. (1) enables to measure the thermal integral from hourly average temperatures, improving the prediction scale of the biological process[61]. MGDD also provides an excellent metric to link $Xf_{PD}$ growth in culture with PD development as, once the pathogen is injected into the healthy vine, symptoms progression mainly depends upon the bacterial load (i.e., multiplication) and the

movement through the xylem vessel network, which are fundamentally temperature-dependent processes[38,62]. Moreover, MGDD can be mathematically related to the exponential or logistic growth of the pathogen within the plant (Supplementary Note 2B).

Interannual infection survival in grapevines plays a relevant role when modelling PD epidemiology. In our model, we assumed a threshold of five or more symptomatic leaves for these chronic infections based on the relationship between the timing and severity of the infection during the growing season and the likelihood of winter recovery[38,39,42]. This five-leaf cut-off was grounded on: (i) the bimodal distribution in the frequency of the number of symptomatic leaves among the population of inoculated grapevines (Supplementary Fig. S1), whereby vines that generally show less than five symptomatic leaves at 12 weeks after inoculation remain so in the following weeks, while those that pass that threshold continue to produce symptomatic leaves, and (ii) the observed correlation between the acropetal and basipetal movement of Xf along the cane (Supplementary Fig. S1). The likelihood of developing chronic infections as a function of accumulated MGDD among the population of grapevine varieties was modelled using survival analysis with data fitted to a logistic distribution $\mathcal{F}(MGDD)$. A minimum window of MGDD $= 528$ was needed to develop chronic infections (var. Tempranillo), about 975 for a median estimate, while a cumulative MGDD $> 1159$ indicate over 90% probability within a growing season (red curve in Fig. 1c and "Methods").

Next, we intended to model the probability of disease recovery by exposure to cold temperatures. Previous works had specifically

modelled cold curing on Pinot Noir and Cabernet Sauvignon varieties in California as the effect of temperature and duration[39] by assuming a progressive elimination of the bacterial load with cold temperatures[42]. In the absence of appropriate empirical data to formulate a general average pattern of winter curing among grapevine varieties, we combined the approach of Lieth et al.[39] and the empirical observations of Purcell on the distribution of PD in the US related to the average minimum temperature of the coldest month, $T_{min}$, isolines[41]. To consider the accumulation of cold units in an analogy of the MGDD, we searched for a general correlation between $T_{min}$ and the cold degree days (CDDs) with base temperature = 6 °C (see "Methods"). We found an exponential relation, CDD $\sim 230 \exp(-0.26 \cdot T_{min})$, where specifically, CDD $\gtrsim 306$ correspond to $T_{min} < -1.1$ °C (Fig. 1b). To transform this exponential relationship to a probabilistic function analogous to $\mathcal{F}(\text{MGDD})$, hereafter denoted $\mathcal{G}(\text{CDD})$, ranging between 0 and 1, we considered the sigmoidal family of functions $f(x) = \frac{A}{B + x^C}$ with $A = 9 \times 10^6$, $B = A$ and $C = 3$ (Fig. 1c), fulfilling the limit $\mathcal{G}(\text{CDD} = 0) = 1$, i.e., no winter curing when no cold accumulated, and a conservative 75% of the infected plants recovered at $T_{min} = -1.1$ °C instead of 100% to reflect uncertainties on the effect of winter curing.

**MGDD/CDD distribution maps**. MGDD were used to compute annual risk maps of developing PD during summer for the period 1981–2019 (see "Methods"). The resulting averaged map identifies all known areas with a recent history of severe PD in the US corresponding to $\mathcal{F}(\text{MGDD}) > 90\%$ (i.e., high-risk), such as the Gulf Coast states (Texas, Alabama, Mississippi, Louisiana, Florida), Georgia and Southern California sites (e.g., Temecula Valley) (Fig. 2a), while captures areas with a steep gradation of disease endemicity in the north coast of California ($\mathcal{F}(\text{MGDD} > 50\%)$). Overall, more than 95% of confirmed PD sites ($n = 155$) in the US (Supplementary Data 2) fall in grid cells with $\mathcal{F}(\text{MGDD}) > 50\%$.

The average MGDD-projected map for Europe during 1981–2019 spots a high risk for the coast, islands and major river valleys of the Mediterranean Basin, southern Spain, the Atlantic coast from Gibraltar to Oporto, and continental areas of central and southeast Europe (Fig. 2b). Of these, however, only some Mediterranean islands, such as Cyprus and Crete, show $\mathcal{F}(\text{MGDD}) > 99\%$ comparable to areas with high disease incidence in the Gulf Coast states of the US and California. Almost all the Atlantic coast from Oporto (Portugal) to Denmark are below suitable MGDD, with an important exception in the Garonne river basin in France (Bordeaux Area) with low to moderate MGDD (Fig. 2b).

Figure 2a shows how the area with high-risk MGDD values extends further north of the current known PD distribution in the southeastern US, suggesting that winter temperatures limit the expansion of PD northwards[9]. A comparison between MGDD and CDD maps (Fig. 2a vs. Fig. 2c, Fig. 2e) further supports the idea that winter curing is restricting PD northward migration from the southeastern US. However, consistent with growing concern among Midwest states winegrowers on PD northward migration led by climate change[63], we found a mean increase of 0.12% $y^{-1}$ in the areal extent with CDD < 306 ($\sim T_{min} < -1.1$ °C) since 1981, comprising land areas between 103°W and 70°W of the US (Supplementary Fig. S4). Such an upward trend corresponds to 5090 km$^2$ $y^{-1}$ in the potential northward expansion of PD due to climate change and an accumulation of ~193420 km$^2$ of new areas at risk since 1981.

High-CDD values would also be expected to restrict the potential PD colonisation in continental Europe (Fig. 2d). Unlike North America, the East-West distribution of major European mountain ranges together with the warming effect of the Gulf

Stream decreases the likelihood of cold winter spells reaching the western Mediterranean coast. $\mathcal{G}(\text{CDD})$ between 100% and 95% (i.e., recovery probability <5% – low winter curing) are mostly prevalent below 40°N latitude in the southwest Iberian Peninsula and Mediterranean islands and coastlands (<50 km away). Above 40°N latitudes, CDD < 100 are encountered mainly in the Atlantic coast and Mediterranean coast and islands (Fig. 2d). In contrast, central and southeast Europe show high CDD values likely preventing $Xf_{PD}$ winter survival on infected grapevines.

In Fig. 2e, f, we show the average climatic suitability for PD establishment only from the mechanistic relation between $Xf_{PD}$ and temperature. Although all areas with current $Xf_{PD}$-related outbreaks are identified, risk predictions based only on the combination of MGDD and CDD could lead to overestimations, as this approach overlooks disease transmission dynamics and climate interannual variability.

**PD global risk**. We ran several simulations of the model Eq. (7) with $R_0$ values between 1 and 14 to validate PD spatiotemporal distribution in the US. We found $R_0 = 8$ as the optimal parameter for maximising the area under a ROC curve (Supplementary Fig. S5), returning an accuracy of more than 80%, except for 2006, due to data obtained from an area at the transient-risk zone (Supplementary Fig. S7 and Table 1). For Europe and the rest of the world, we derived a $R_0 = 5$, as a maximal baseline estimate for modelling PD transmission (see "Methods" and Supplementary Note 2D). These $R_0$ values should be taken as operating estimates for the model. From the model simulations Eq. (7), we obtained a risk index $r$ that measures the relative exponential growth rate in the population of infected plants at the epidemic onset with respect to the maximum growth, $r = 1$. This index served to rank the epidemic-risk zones in high (>0.9), moderate (0.66–0.9), low (0.33–0.66), and very low (~0.075–0.33) risks (see Fig. 1f, "Methods", and Supplementary Note 2E).

To date, PD is mainly restricted to the American continent with some unrelated introductions of $Xf_{PD}$ to Taiwan and Majorca (Spain) from the United States[12,13]. To assess the risk of PD establishment elsewhere, we projected our epidemiological model into the main winegrowing regions of the Northern Hemisphere (US, Europe, and China) and Southern Hemisphere (Chile, Argentina, South Africa, Australia, and New Zealand) (Fig. 3a–e). We found that emerging wine-producing areas in China are predominantly located in non-risk zones, whereas only some vineyards in the Henan and Yunnan provinces fall in transition and moderate-high risk zones (Fig. 3b and Supplementary Data 3). In Europe, 92.1% of the territory is in non-risk zones and 6.1% is included in the epidemic-risk zone, with only 1.9% showing a high-risk index and 1.5% a moderate risk (Supplementary Table S2). The model also reveals a progressive transition from areas without risk ($r(t) < 0$) before 1990 to epidemic-risk zones with low-risk indexes by 2019[57] (see Movies), mainly affecting the basins of the rivers Po in Italy, Garonne, and Rhone in France and Douro/Duero in Portugal and Spain. This represents a mean increase of 0.21% $y^{-1}$ in the epidemic-risk zone, a rate 3.5-times greater than that of the eastern US, which could increase the likelihood of PD establishment in Europe in the coming decades. In the US, most states around the Gulf Coast show high-risk indexes, whereas, around 37.5% of California's surface is suitable for epidemics with high growth rate incidence (Supplementary Table S3).

In the Southern Hemisphere, vineyards at non-risk or transient epidemic-risk zones predominate—e.g., non-risk in New Zealand and Tasmania (Fig. 3c). Risk indexes in areas where PD can become established ($r(t) > 0$) range from very low to low for most coastal vineyards in Australia (west, south and east) with

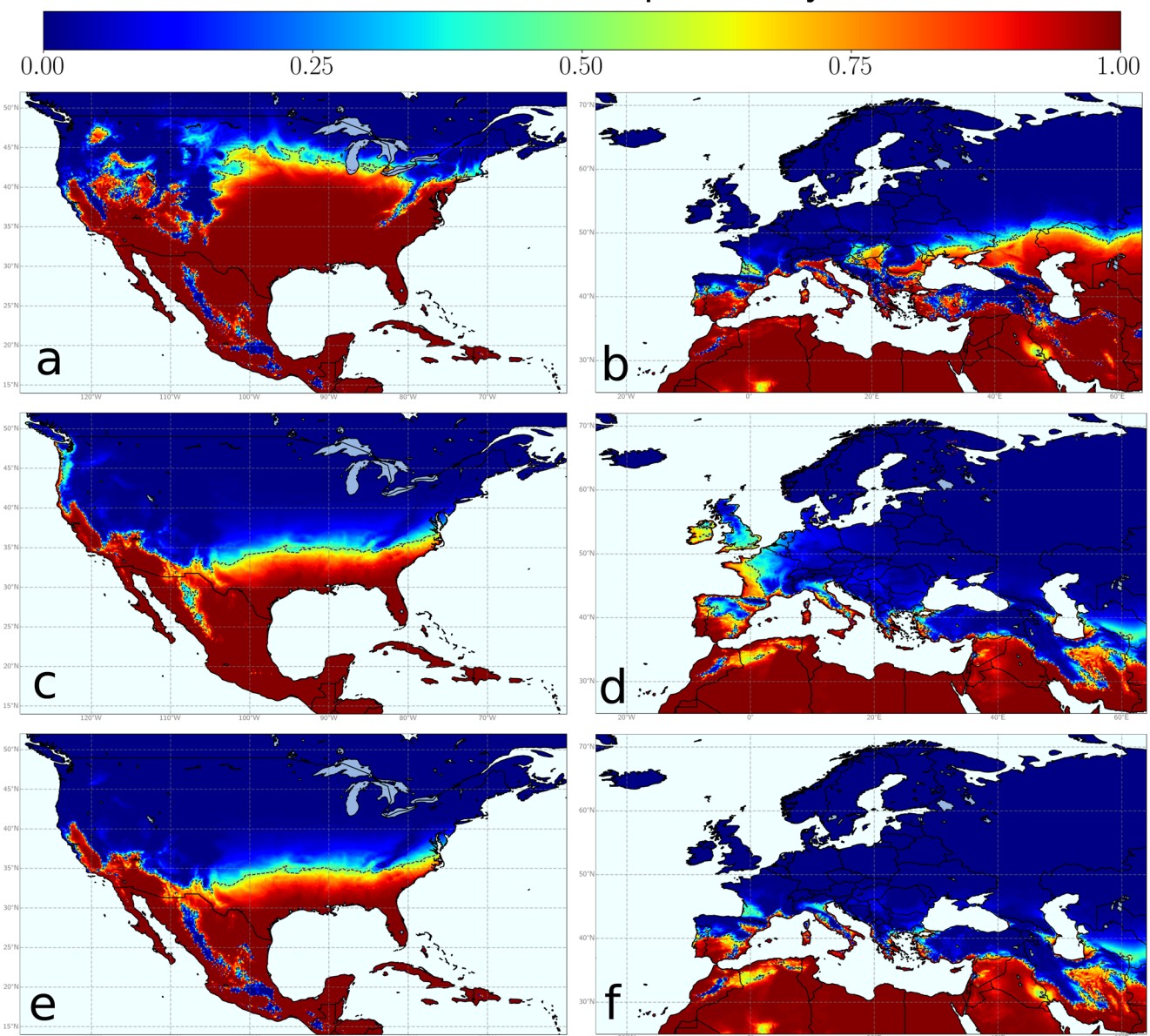

**Fig. 2 Average thermal-dependent maps for Pierce's disease (PD) development and recovery in North America and Europe.** PD development during the growing season based on average $\mathcal{F}$(MGDD) estimations between 1981 and 2019 in North America (**a**) and Europe (**b**) derived from the results of the inoculation experiments on 36 grapevine varieties. Large differences in the areal extension with favourable MGDDs can be observed between the US and Europe. The winter curing effect is reflected in the distribution of the average $\mathcal{G}$(CDD) for the 1981–2019 period in the United States (**c**) and Europe (**d**). A snapshot of the temperature-driven probability of chronic infection averaged for the 1981–2019 period is obtained from the joint effect of MGDD and CDD in North America (**e**) and Europe (**f**). Warmer colours indicate more favourable conditions for chronic PD and the dashed line highlights the threshold of chronic infection probability being 0.5.

**Table 1 Validation of model predictions.**

| Year | Presence | Absence | TP | TN | Accuracy |
|------|----------|---------|----|----|----------|
| 2001 | 16 | 5 | 15 | 3 | 86% |
| 2002 | 12 | 2 | 11 | 1 | 86% |
| 2005 | 4 | 2 | 4 | 1 | 83% |
| 2006 | 8 | 0 | 4 | 0 | 50% |
| 2015 | 53 | 0 | 51 | 0 | 96% |
| TOTAL | 93 | 9 | 85 | 5 | 88% |

The items are locations where PD was present or absent. TP corresponds to true positives and TN to true negatives according to our model with $R_0 = 8$.

somehow more suitable conditions in the interior of New South Wales, Greater Perth and Queensland (Fig. 3c); a general very-low or low-risk indexes are predicted in the Western Cape in South Africa (Fig. 3d); overall very-low but localised low to moderate risk indexes in some areas in Chile; and low to moderate growth of the number of infected vines in most of Argentina, being this the wine-growing country with the highest risk (Fig. 3a). Detailed information on areas with non-risk, transient risk and risk indexes (i.e., disease-incidence growth rates) in areas with the potential risk of establishment by country and regions is provided in Supplementary Table S4 and Supplementary Data 3.

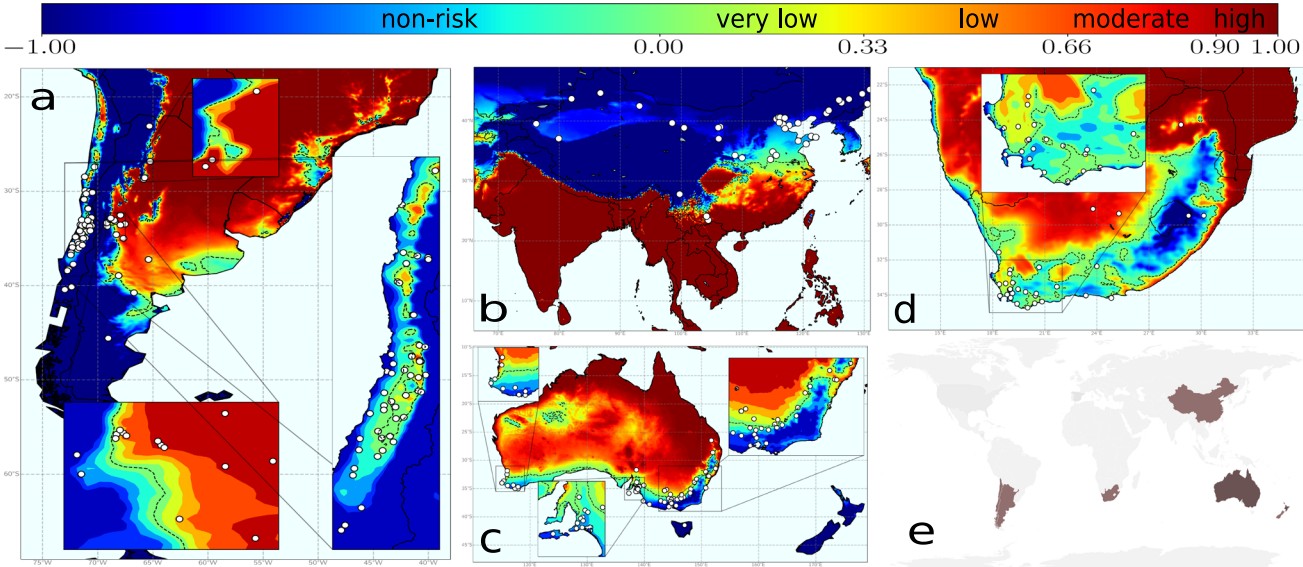

**Fig. 3 Climate-driven risk maps for PD establishment in main viticulture regions worldwide under a baseline $R_0 = 5$ scenario.** White dots indicate the main vineyard areas in the wine-growing regions of China and the Southern Hemisphere. **a** Chile and Argentina; **b** Asia with special attention to China; **c** Australia and New Zealand (wine areas are not marked as the whole country is without risk); and **d** South Africa. **e** Global distribution of main wine-producing areas analysed. The risk index $r_j(t)$, express the relative exponential growth rate of the disease incidence, and was scaled from 0.1 to 1 and ranked as very low (0.10–0.33), low (0.33–0.66), moderate (0.66–0.90) and high (>0.90).

Risk indexes may vary within epidemic-risk zones if any of the epidemiological parameters governing transmission change. As expected, $I(t) < I(0)$ boundaries increasingly displace to northern latitudes in the US and Europe under higher transmission scenarios, increasing the risk-epidemic zones significantly (Fig. 4a–f). The line representing the outbreak extinction i.e., the non-risk zone $r(t) < -0.09$, in the validated $R_0 = 8$ scenario for the US, falls at some distance above the isoline $T_{min} = -1.1\,^\circ$C in comparison to the $R_0 = 5$ scenario (Fig. 4c vs Fig. 4a and ref. [57], Movies). This distribution pattern holds and moves slightly northward over time in parallel to global warming, although the trend of PD latitudinal change is moderated by high-CDD values (i.e., cold accumulation). In addition, the disease extension also declines due to CDD interannual fluctuations in the simulations. Cold waves periodically occur that reach latitudes close to the Gulf, such as those that occurred in 1983, 1993, 1995, 2000, 2009, and 2013[57] (see Movies), thus preventing PD expansion northward. The magnitude of this decrease is revealed after comparing the average annual increase of the areas between $r(t) > 0$ and CDD < 306 lines. From 1981 to 2019, the area with risk $r(t) > 0$ increased at a rate of 0.05% y$^{-1}$, while that of CDD < 306 by 0.12% y$^{-1}$, an important difference not explained alone by CDDs without considering climate fluctuations (Supplementary Fig. S4).

We checked whether using a beta function produces changes in the risk indexes with respect to the Arrhenius-based approach. Firstly, we needed to calibrate the model using the probability of developing chronic infections, as in Fig. 1c, with the values of MGDD obtained with the beta function. We found little differences, mainly a decrease in risk index in the transition zones between risk and non-risk zones ((Supplementary Fig. S12) and (Supplementary Fig. S13)), and non-significant differences in risk zones at the global scale.

**PD risk projections for 2050.** Global shifts in the risk index $r_j(t)$ between 2019 and those projected for 2050 were calculated under the same baseline scenario (Fig. 5a–f, "Methods"). Our simulation shows a generalised increasing trend mainly due to shifts from transition zones to epidemic-risk zones with very low or low-risk

indexes in the main wine-growing regions, except for the US. Here the epidemic-risk zone would increase by 12.8% with the greater increments in the high-risk index category (22.7%) and a decrease in the transition zones (Supplementary Table S5). Much less surface would be included in the epidemic-risk zone in Europe (8.6%) compared to the US (36.5%). However, the epidemic-risk zone would expand by 40.0% with respect to 2020, a rate more than three times higher than that of the US (Supplementary Table S6). Such increases are due to the emergence of previously unaffected areas in 2020 evolving into epidemic-risk zones by 2050, and epidemic growth-rate increases in already epidemic-risk zones in 12 of 42 countries (Supplementary Table S2). Among these 12 countries, however, there is substantial variation in the risk index increments within epidemic-risk zones with respect to 2019 (Supplementary Table S6). While non-risk zones still cover 87.6% of Europe's land area, epidemic-risk zones with high-risk indexes are expected to be almost two-fold higher than that of 2019, comprising 3.2% of Europe (Table 2).

**Risk based on vector information.** So far, we have ignored the distribution of known and potential vector species due to their large number in the Americas and the limited quantitative information generally available. In the case of Europe, given *P. spumarius* prevalence as a potential vector and its wide distribution, we added a vector layer in a spatially dependent $R_0(j) = R_0^{max} v(j)$, where $v(j)$ is the climatic suitability for the vector ("Methods"), $v = 1$ implies optimal climatic conditions with no constraints for the vector population size, while $v = 0$ implies unsuitable climatic conditions and its absence (Supplementary Fig. S8). According to the model, no European zone shows a high-risk index and barely 0.34% of the territory falls in areas with potential moderate exponential growth rates in disease incidence (Supplementary Table S7). Irrespective of vineyard distribution, we estimated that PD could potentially become established (i.e., $r(t) > 0$) at a maximum of 3.1% of the territory, while the area at moderate-risk index would be 5-times lesser than the model without the vector's climate suitability layer, this latter more in consonance with other proposed risk maps[45,46].

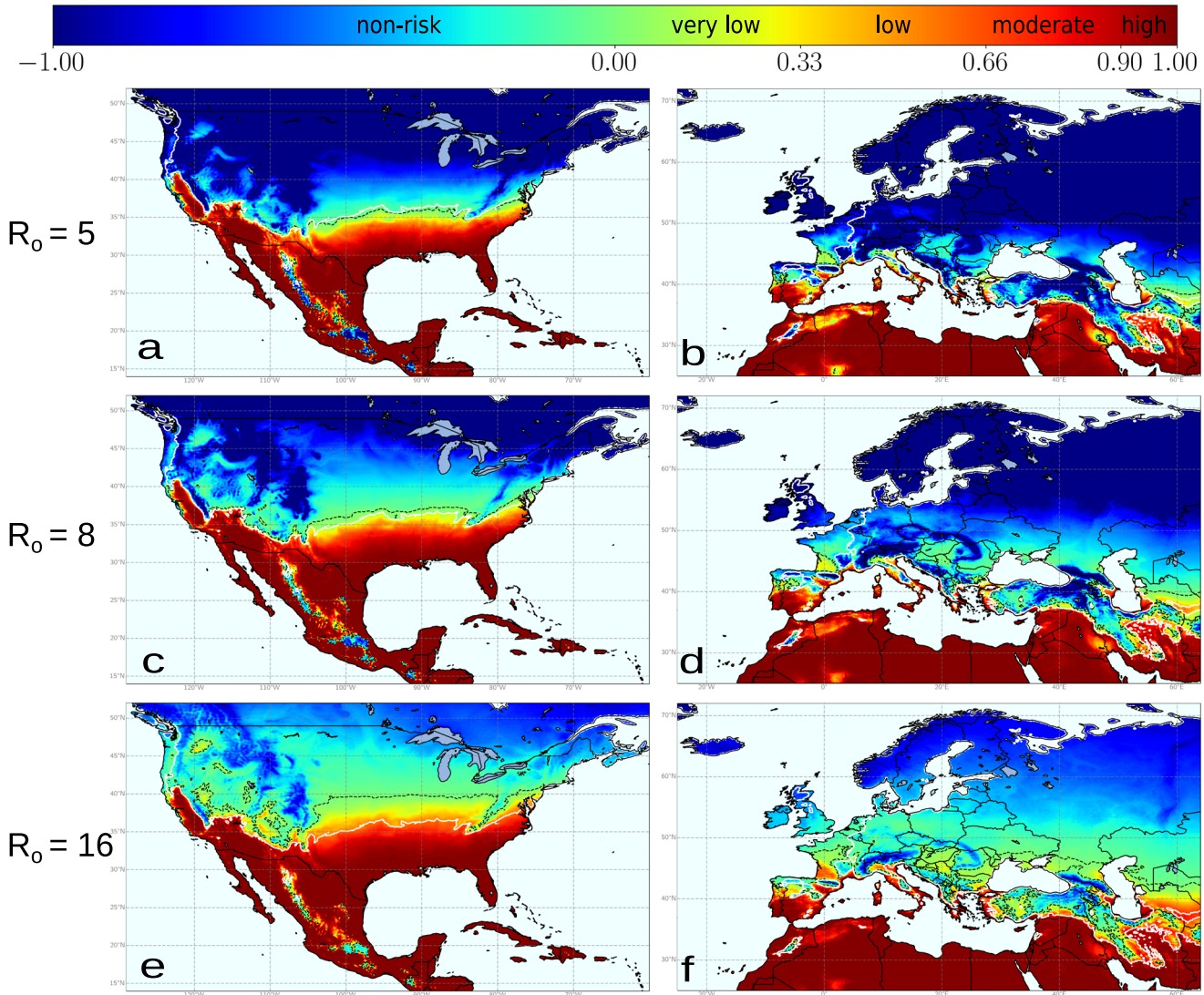

**Fig. 4 Temperature-driven dynamic-model simulations for PD establishment from 1981 to 2019 under different $R_O$ scenarios with a spatially homogeneous vector distribution.** For comparison, the baseline scenario with a $R_O = 5$ for Europe is projected to North America (**a**) capturing to some extent the distribution and severity of PD in that continent. In Europe (**b**) high-risk areas (i.e., $r_j(t) > 0.90$) are restricted to the coastal Mediterranean and the south of the Iberian Peninsula; black dash line separate areas with $r(t) > 0$ where theoretically PD can thrive. Under higher $R_O$ scenarios, $R_O = 8$ for North America (**c**) and Europe (**d**), the dash lines tend to separate from isoline $T_{min} = -1.1\,°C$ (white line); and even more in extreme transmission pressure $R_O = 16$ for North America (**e**) and Europe (**f**).

Such differences in the projected risks are mainly concentrated in the warmest and driest Mediterranean regions and are due to uncertainties concerning temperature-humidity interactions in the ecology of the vector[35].

**Combining vineyard land cover across Europe with the model output**. When we integrate into the model a layer of vineyard surface from Corine-Land-Cover, we find that PD could potentially become established (i.e., $r(t) > 0.075$) in 22.3% of the vineyards in Europe. However, no vineyard is in epidemic-risk zones with a high-risk index and only 2.9% of the vineyard surface is at moderate risk (Supplementary Table S8). The areas with the highest risk index ($r(t)$ between 0.70 and 0.88) are mainly located in the Mediterranean islands of Crete, Cyprus and the Balearic Islands or at pronounced peninsulas like Apulia (Italy) and Peloponnese (Greece) in the continent (Fig. 6a and Supplementary Table S8). Most vineyards are in non-risk zones (42.1%), whereas 35.6% are located in transition zones with presently non-risk but where Xf$_{PD}$ could become established in the next decades

causing some sporadic outbreaks. In Supplementary Data 4 and Supplementary Table S8, we provide full details of the total vineyard areas currently at risk for each country and region.

Our model with climate and vector distribution projections for 2050 indicates a 55.8% increase in the epidemic-risk zone in Europe (Fig. 6b). This increment would be mainly due to the extension of epidemic-risk zones with very low and low-risk indexes. However, within the epidemic-risk zones, areas with moderate risk indexes would decrease from 114925 ha in 2020 to 43114 ha in 2050, and no vineyards would be at high risk (Fig. 6b; see Supplementary Table S9 and Supplementary Data 4). Counterintuitively, our model indicates a substantial increase in the area where PD could establish and become endemic for 2050, but a moderate decline in those areas where crop damage could be expected to be significant (e.g., Balearic Islands, Crete, Cyprus, Apulia).

**Discussion**

We introduce an epidemiological approach to assess the risk of PD establishment and epidemics in vineyards worldwide. The

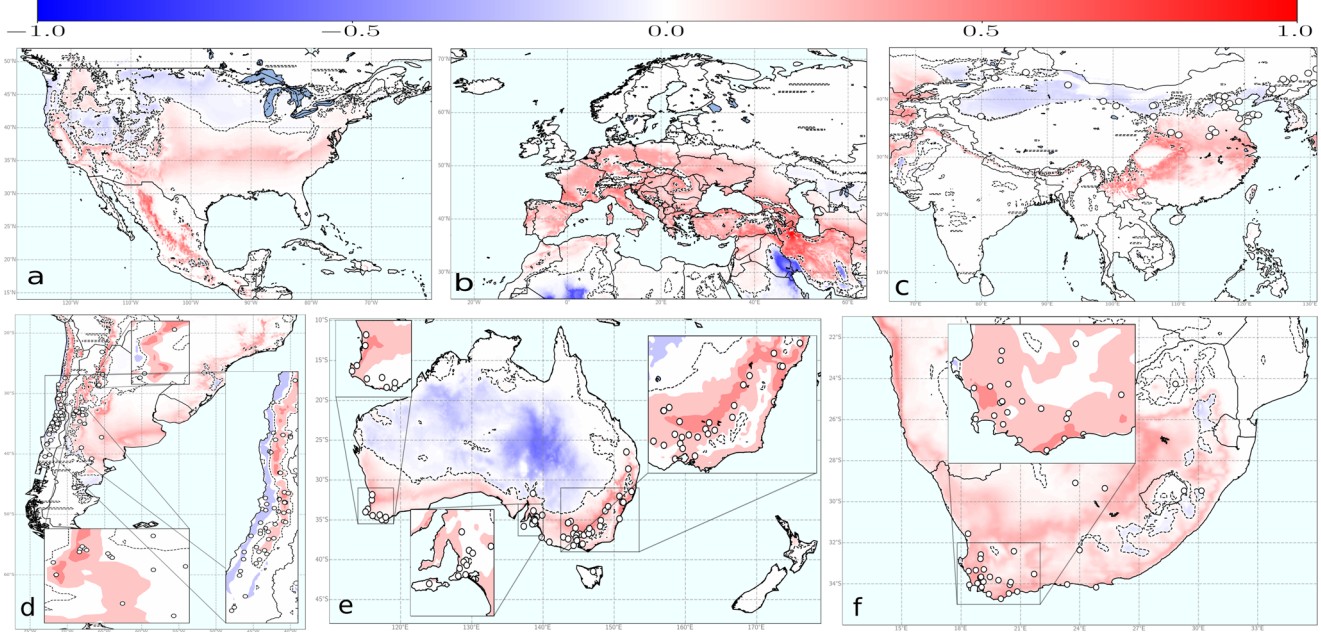

**Fig. 5 Global shifts in PD risk index ($r_j(t)$) from 2020 to 2050.** To build the maps, we have assumed a spatial homogeneous vector distribution and a $R_O = 5$ scenario, except for the US where a $R_O = 8$ has been used in the model simulations. **a** North America; **b** Europe; **c** Asia; **d** South America; **e** Australia and New Zealand; and **f** South Africa. Risk-index increases are in red and decreases are in blue. The dashed line represents the spatial threshold where $r_j(t)$ difference changes from negative to positive.

**Table 2 Shifts in risk areas for Pierce's disease in Europe projected for 2050 under a $R_O = 5$ scenario.**

| Risk | 2050 km² | 2020 km² | Difference km² | Difference (%) | 2050 (%) Area | 2020 (%) Area |
|---|---|---|---|---|---|---|
| No risk | 8885300.5 | 9334178.7 | −448878.2 | −4.8 | 87.6 | 92.1 |
| Transition | 381081.3 | 182872.6 | 198208.7 | 108.3 | 3.8 | 1.8 |
| Very low | 189025.3 | 179225.7 | 9799.6 | 5.5 | 1.9 | 1.8 |
| Low | 207599.4 | 104143.1 | 103456.3 | 99.3 | 2.1 | 1.0 |
| Moderate | 154780.5 | 148621.4 | 6159.0 | 4.1 | 1.5 | 1.5 |
| High | 322225.9 | 190971.4 | 131254.5 | 68.7 | 3.2 | 1.9 |

The model was run assuming the same homogeneous spatial distribution of the vector for the whole period.

model includes the dynamics of the infected-host population, which enables estimating the initial exponential growth/decrease rate of the disease incidence. Unlike SDM correlative studies, Bayesian or, in general, machine learning black-box approaches, our model goes beyond by providing a mechanistic framework and thus explanatory power. In addition, it is flexible enough to simulate different climate and transmission scenarios, allowing, for instance, the incorporation of information on the spatial distribution of the vector. Comprehensive global PD risk maps result from the model simulations with historical climatic data. A web page is included, showing simulations with different parameters to estimate the risk of PD anywhere[57].

Temperature regulates key physiological processes of the ectothermic organisms involved in PD and thus limits the thermal range in which they can thrive[52]. $Xf_{PD}$ multiplication and survival within vine xylem vessels not only characterise PD, but also determine the bacterial population dynamics[38,62]. PD symptom development can be therefore characterised as a thermal-dependent continuous process within the range of $Xf_{PD}$'s cardinal temperatures[53]. The combination of MGDD metrics with robust experimental data provides a reliable predictor of climatic suitability and the probability of developing PD during the summer, whereas CDD accounts for the effect of cold-

temperature exposure on infected-plant recovery. This opposite contribution of MGDD and CDD in the demography of infected plants shapes the impact of climate variability on the epidemic dynamics in the early stages of the invasion (Fig. 1d). Given that the physiological basis of the plant-Xf interaction leading to symptoms development is poorly understood, we caution that other environmental factors, such as drought, nutrient status or crop management may modulate symptom expression and hence add an error in the MGDD parameter not measured in this work. Nonetheless, we deem the error range would be smaller than the differences in the accumulated MGDDs needed to reach the same disease level among varieties (i.e., regional differences) and smaller than the interannual MGDD oscillations found in most locations. In addition, our model is general enough to allow for other functions or adjustments of the relationship between $Xf_{PD}$'s growth rate and temperature in vitro as better experimental data become available. However, we deem that the differences in the risk indices would vary very little in risk zones, as we observe in PD risk maps for Europe when a Beta function is applied instead of the Arrhenius-based approach to adjust the MGDD (Supplementary Figs. S12 and S13).

Knowledge of insect distribution is crucial for predicting epidemic outbreaks of endemic diseases, as well as the risk of

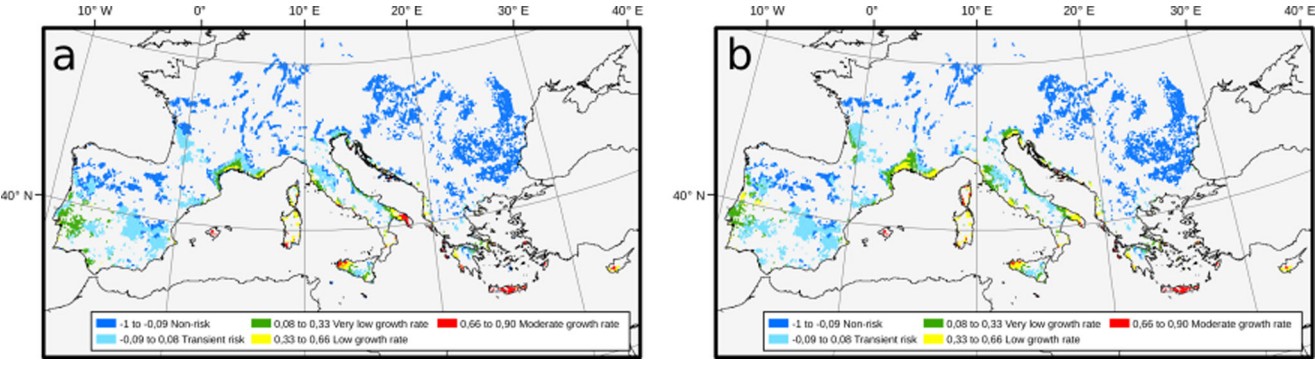

**Fig. 6 Intersection between Corine-land-cover vineyard distribution map and PD-risk maps for 2020 and 2050.** Data were obtained from Corine-land-cover (2018) and the layer of climatic suitability for *P. spumarius* in Europe from[35]. The surface of the vineyard contour has been enlarged to improve the visualisation of the risk zones and disease-incidence growth-rate ranks. **a** PD risk map for 2019 and its projection for 2050 (**b**). Blue colours represent non-risk zones and transient risk zones for chronic PD ($R_0 < 1$). The 2050 map shows some contraction of epidemic-risk zones with moderate risk indexes in Mediterranean islands and Apulia (Italy) as the climate becomes hotter and dryer.

invasion by emerging vector-borne pathogens[49,56,64]. Given the great diversity of known and potential vectors that can transmit PD[30], it has not been possible to include each region's particular vectors in the model. Therefore, when evaluating the risk of PD on a global scale, we have considered a homogeneous spatial distribution of the vector (fixed $R_0$), except in Europe where there is information on the main vector (Supplementary Fig. S8). As expected, the European case shows how models that assume a homogeneous spatial distribution of the vector generally produce epidemic risk zones with higher risk indexes than models that include a heterogeneous spatial distribution (Supplementary Table S2 vs. Supplementary Table S7). This lack of information about vectors is one of the main reasons why the risk of vector-borne plant diseases is often overestimated.

Risk overestimations may involuntarily stem from other additional sources too. Using mean data as inputs in epidemiological models can lead to biased results when response functions are nonlinear and climate variability is not accounted for[53]. This study presents experimental evidence of a non-linear relationship between MGDDs and PD chronic infections and indirect empirical evidence of a non-linear relationship between CDDs and PD recovery (Supplementary Fig. S9). Such a non-linear response consequently greatly impacts reducing the risk of PD establishment and steeping the spatial gradients in risk maps (Figs. 4 and 6). Moreover, MGDDs and CDDs might help to explain why disease pressure is much higher in the southeastern US than in California and Europe (Figs. 2 and 4) or, for example, earlier reports of PD outbreaks in Kosovo[65]. Cooler summer nights in California and a shorter growing season compared to those found in the Gulf states in the southeastern US explain the difference in the accumulated MGDD for both areas. In the case of Kosovo, CDD values above certain thresholds could have led to the extinction of incipient outbreaks driven by several years with MGDD in the conducive range of PD (Fig. 2).

Our PD risk map for Europe confirms previous predictions for the subsp. *fastidiosa* from SDMs[45]. Both approaches make congruent predictions on PD potential distribution, providing convergent lines of independent evidence of climate suitability. However, our risk maps go further by incorporating in the epidemic-risk zones information on the relative exponential growth rates in the potential disease incidence. In general terms, the epidemic-risk map including vector information indicates a low risk for chronic PD. Only ~ 0.34% of European vineyard surface, mainly located in Cyprus, Crete, Sardinia, part of Sicily and the Balearic Islands, meet climatic conditions for PD to become endemic and cause significant damage (Supplementary

Table S7 and Supplementary Data 4). Other regions such as Bordeaux, Portugal, Rhône Valley, and the Veneto region, would be included in epidemic-risk zones but with very low to low exponential growth rates in disease incidence. By contrast, notorious wine-growing regions in Spain (e.g., Rioja, Ribera del Duero), France (e.g., Burgundy) and Italy (e.g., Piedmont) currently fall within areas considered as non-risk zones, transient-epidemic zones or epidemic-risk zones with very-low risk indexes (Fig. 6).

The dynamic nature of the simulation outputs already points to a progressive global increase in the areal extension of PD epidemic-risk zones ($r(t) > 0$) in the last decade, irrespective of vineyard distribution (see movies on ref. [57]). This is even more accentuated in the model projections for 2050, which point out a global expansion of PD epidemic-risk zones at different velocities among continents due to climate change (Fig. 5). For example, many important viticulture areas in western Europe included in non-risk or transition zones before 1990 are progressively shifting to hotter summers and milder winters and hence would be increasingly suitable for the disease within the extrapolated current scenario. This is further illustrated by a 40% increase of the potential epidemic-risk zone by 2050 concerning 2020 for Europe and more moderate increases in the United States and the Southern Hemisphere (Fig. 5). Nonetheless, our model projection for 2050 that includes spatial heterogeneity in the vector distribution, as in Europe, would indicate lower transmissibility because global change is predicted to have negative effects on *P. spumarius* abundance in Europe[35,66]. At the global scale, there is certainly scientific consensus that climate change will follow a general pattern summarised in the paradigm "dry gets drier, wet gets wetter"[67]. In agreement, our model projection for PD on the vineyards of Majorca (Spain) suggests shifts to slightly less favourable conditions for $Xf_{PD}$ transmission and an expected progressive decrease in the impact of the disease by 2050. This example and others in Mediterranean islands (see Supplementary Data 4) advocate for certain caution when interpreting climate change projections, especially in other Mediterranean climates of the world, where the complex interactions between humidity and temperature can limit the presence and abundance of vectors (Supplementary Fig. S8).

The scope of our study excludes location-specific complexities surrounding PD ecology due to scale limitations. The spatial distribution of the vector is considered only for the V. vinifera-$Xf_{PD}$-P. spumarius pathosystem in Europe, so $R_0$ estimations could locally differ in other wine-producing regions elsewhere (Fig. 3). Disease incidence thus could locally vary where the

climate is conducive to PD. Such variation is because transmission rates tend to increase exponentially rather than linearly under environmental conditions favouring vector abundance[43], as has been observed at a local scale on vineyards of Majorca[12]. Our study also does not contemplate likely changes within the PD pathosystem. To date, PD is caused by $Xf_{PD}$ (i.e., ST1/ST2), but other genotypes of the subsp. *fastidiosa* or other subspecies and their recombinations could arise in the future with different ecological and virulence traits[19]. On the other hand, new vector species could be accidentally brought in[30], as exemplified with the introduction of the glassy-winged sharpshooter (*Homalodisca vitripennis*) in California, modifying transmission rates and disease incidence in new areas[44]. To capture these uncertainties in relation to the vector, we have performed simulations with $R_0 = 8$ and $R_0 = 16$ (Fig. 4). Remarkably, a comparison of PD risk maps for Europe with different $R_0$ suggests for non-Mediterranean areas the need to stress more surveillance on the introduction of alien vectors rather than in the pathogen itself. This is because, under the current scenario ($R_0 = 5$) with *P. spumarius* as the main vector, most of the non-Mediterranean vineyards would not support the establishment of PD, but the introduction of new insect vectors with greater transmission efficiency ($R_0 = 8$) could compensate for the climatic layer and increase the risk index above 0. In addition, differences in grapevine varietal response alongside virulence variation among Xf strains may slightly modify PD thermal tolerance limits and therefore locally modulate epidemic intensity (see details in Supplementary Information). Such an effect could be seen with cv. Tempranillo, a widely planted variety in northern Spain (Supplementary Table S1); the rate of symptom progress and systemic movement is higher than the average varietal response to $Xf_{PD}$ (i.e., lower MGDD), which in addition might imply higher survival rates. This point calls for further testing in the field.

Our model partially explains why PD has not become established in continental Europe and other main wine-growing regions worldwide during the last 150 years, in contrast to other exotic diseases and pests brought in with native vines from the US[5–8]. We suggest that the underlying causes of this low-invasiveness risk in Europe are fundamentally two: (i) low climatic suitability for chronic PD and (ii) a climatic mismatch between environment conditions suitable for both the vector and the pathogen and their interplay in disease dynamics, similar to the situation recently described for the *V. vinifera*-$Xf_{PD}$-*P. spumarius* pathosystem in northern California[33]. Currently, suitable conditions for the pathogen's invasion mostly concur in Mediterranean islands and coastlands (Supplementary Data 4). Likewise, similar results would be expected in other Mediterranean climates of the main winegrowing regions of the Southern Hemisphere if a vector spatial distribution layer is incorporated in the model simulations (see ref. [57]). Finally, although increasing global warming will extend epidemic-risk zones in all continents, some caution is recommended to not incur risk overestimation, as we show in the PD risk projections for 2050 in Europe when taking into account the vector spatial distribution; complex interactions between temperature and humidity in the ecology of the vectors may have a great effect in their distribution, abundance and thus transmission capacity[35]. There is an urgent need to fill the knowledge gap on the ecophysiology for each potential vector to downscale PD model predictions to local and regional situations.

## Methods

**Inoculation tests**. $Xf_{PD}$-inoculation tests were conducted in 2018, 2019, and 2020. A sample of 36 local, regional and international wine-grape varieties was selected, which included nine of the 10 most cultivated wine-grape varieties representing

more than 80% of the worldwide vineyard surface (https://www.oiv.int). Plants were randomly distributed in 12-plant rows along an insect-proof net tunnel and exposed to environmental temperature. In total, 57 rootstock-scion combinations were pin-prick mechanically inoculated[25] with two strains of Xf. subsp. *fastidiosa* (ST1) isolated from grapevines in Majorca. Disease severity was rated by counting the number of symptomatic leaves eight weeks after inoculation in mid-May and then every two weeks until the 16th week[12]. Full details on the inoculation conditions, isolates, disease score, and statistical analysis are provided in Supplementary Information, Supplementary Table S8, and Supplementary Data 1.

**Modified Growing Degree Days**. We generalised McMaster & Wilhelm's[58] formulation of growing-degree days to account for the growth rate of $Xf_{PD}$ as a function of temperature under optimal culture conditions based on the well-known Arrhenius law valid in the relevant temperature range for Xf (Supplementary Note 2A). Specific growth rate ($k$) values at different temperatures were extracted from the publication of Feil & Purcell[38] to build the mathematical function $f(T)$ describing the Xf's instantaneous growth rate dependence on temperature according to

$$f(T) = \begin{cases} 0 & \text{if } T < T_{base} \\ m_1 \cdot T - b_1 & \text{if } T_{base} \leq T < T_1 \\ m_2 \cdot T + b_2 & \text{if } T_1 \leq T < T_{opt} \\ m_3 + b_3 & \text{if } T_{opt} \leq T < T_2 \\ m_4 + b_4 & \text{if } T_2 \leq T_{max} \\ 0 & \text{if } T \geq T_{max} \end{cases}$$

where $T_{base} = 12°C$, $T_1 = 18$, $T_{opt} = 28°C$, $T_2 = 32$ and $T_{max} = 35°C$; the slopes are $m_1 = 0.66$, $m_2 = 1$, $m_3 = -1.25$ and $m_4 = -3$ and the intercepts are $b_1 = -8$, $b_2 = -14$, $b_3 = 4$ and $b_4 = 105$.

MGDD is then defined as:

$$\text{MGDD}(t) = \frac{1}{24} \sum_{\tau \in t} f(T(\tau)), \tag{1}$$

where $\tau$ is expressed in hours, $t$ in years and we divide by 24 to obtain MGDD($t$) in degree days. To compare whether using other functions for Xf's growth rates in vitro could yield differences in the risk indexes, we also fitted data to a smooth Beta function commonly used to represent the thermal response in biological processes[59,60].

**Disease progress with temperature**. Hourly mean temperature data were recorded between April 1 and October 31 in 2018, 2019, and 2020 with an automated weather station (Quimisur, IQ2000). The temperature sensor was at a two-metre height from the bare ground and around five metres from the entrance of the insect-proof net tunnel. To characterise the progress of PD symptoms, we converted into MGDD units the cumulative hourly mean temperatures measured from the time of inoculation to the day of disease evaluation using Eq. (1). In total, 15 MGDD levels were estimated corresponding to weeks 8, 10, 12, 14, and 16 after inoculation in the years 2018, 2019, and 2020, respectively. Data on the number of symptomatic leaves (severity) for each plant and MGDD levels were pooled in a single database to seek a generalised average thermal response pattern among the population of *V. vinifera* varieties (see Supplementary Data 1). To model the probability of chronic infections (i.e., persistent year-to-year infections), we used a survival analysis, where the event of interest depends on the cumulative MGDD rather than time. First, we defined a chronic infection cut-off point to transform the number of symptomatic leaves into binary data. Previous research had evidenced that early grapevine infections, in addition to producing more extensive and severe PD symptoms, are more likely to survive the following year than late infections[38,39,42]. Furthermore, susceptible cultivars generally show lower recovery percentages compared to the less susceptible ones in the field[68,69]. Similarly, we observed in our inoculation assays that the majority of infections that reach around five or more symptomatic leaves 12 weeks after inoculation continue to develop more symptomatic leaves the following weeks, while for plants that do not exceed that threshold, symptoms tend to remain stagnant. These results indicate a low probability of survival for infections showing few symptomatic leaves during the growing season and thus support our heuristic approach of assigning five or more symptomatic leaves as a threshold for chronic infection (see Supplementary Information and Supplementary Fig. S1 for assumptions of chronic infection). Using the "survival" package in R[70], we analysed the cumulative probability of developing chronic infections as a function of MGDD. $F$(MGDD) was adjusted to the experimental data by the nonlinear least squares method. The 10th, 33rd, 50th, 66th, and 90th percentiles were used to scale the risk of the total MGDD in the logistic function, $\mathcal{F}$(MGDD) (Fig. 1c).

**Disease recovery through winter curing**. We modelled winter curing considering the effect of temperature duration below a threshold temperature, where we assume that the bacterial killing process increases in efficiency with decreasing temperatures[39]. To adjust a probabilistic model to the accumulation of cold units, we took as reference the distribution and severity of PD in the US proposed by Purcell based on the isolines of the mean $T_{min}$ of the coldest month (available in ref. [41]) where PD is rare ($T_{min}$ between −1.1 °C and 1.7 °C), occasional (1.7–4.5 °C)

and severe (>4.5 °C). Noteworthy, the projection of these isolines in Europe has predicted with some precision the distribution of the establishment of Xf in the continent[45]. To capture the accumulation nature of the chilling process at different climatic zones, we determined the global average correlation between $T_{min}$ and the average accumulated CDD between November 1 and March 31 in the northern hemisphere and between April 1 and October 31 in the southern hemisphere using 6,487,200 points distributed throughout the planet. The CDD was estimated as

$$\mathrm{CDD}(t) = \frac{1}{24} \sum_{\tau \in t} (6 - T(\tau)) \text{ for } T_i \leq 6\,^\circ\mathrm{C}, \tag{2}$$

where the threshold 6 °C comes from ref. [39].

**Global climate data, MGDD/CDD computation.** Global mean hourly temperature data were downloaded from the ERA5-Land dataset[71] at 0.1° spatial resolution using GRIB format. The annual average $T_{min}$ of the coldest month was calculated from the hourly average temperature from the ERA5-Land dataset. To calculate the annual MGDD and CDD a simple Julia[72] library was built on top of GRIB.jl package[73]. For the Northern Hemisphere, the accumulated MGDDs were computed from April 1 to October 31, whereas (CDDs) were estimated from November 1 to March 31, and the reverse for the Southern Hemisphere.

**Disease model.** We used a standard susceptible-infectious/infected-recovered (SIR) compartmental model to assess the risk of PD establishment and epidemics worldwide, represented by the following three equations in the large population limit:

$$\begin{aligned} \dot{S} &= -\beta S I / N, \\ \dot{I} &= \beta S I / N - \gamma I, \\ \dot{R} &= \gamma I, \end{aligned} \tag{3}$$

where $S$ is the susceptible host population, $I$ is the infected population, $R$ is the dead population, and $S + I + R = N$ is the total number of vines in the population. The reduction of a vector-borne disease model to a SIR model gives rise to a linear dependence of the basic reproductive number $R_0$ on the vector population (see Supplementary Notes 2F and 4). Vector-plant transmission of the pathogen is approximated with an effective plant-to-plant transmission rate $\beta$ (Supplementary Note 4), as has been done previously for other Xf-related diseases[74], and the transition from the infected compartment to the recovered (dead) compartment is given by the recovery (mortality) rate $\gamma$. In a mean-field approximation of the onset of an outbreak, the basic reproductive number ($R_0 = \beta/\gamma$) defines the exponential growth/decrease stage in the SIR model (Fig. 1e and Supplementary Note 2C). Although the time from infection to vine death depends on the environmental conditions and the grape wine variety, we assigned a mortality rate of $\gamma = 0.2\,\mathrm{y}^{-1}$ based on the estimated median survival time of infected vines in California[25]. The maximum growth rate of the epidemic, relevant for an estimation of the risk of establishment, occurs when $S(t = 0) \sim N$, and was approximated by the (linearised) differential equation,

$$dI/dt \approx \beta I - \gamma I = \gamma I (\beta/\gamma - 1) = \gamma I (R_0 - 1), \tag{4}$$

where we have assumed the initial conditions: $S(t = 0) \approx N$, $I(t = 0) = I(0) \approx 0$ and $R(t = 0) = 0$. This linear differential equation can be integrated exactly:

$$I(t) = I(0) \exp(\gamma (R_0 - 1) t). \tag{5}$$

To account for the effect of temperature in the epidemic process, we modify the previous expression as follows

$$\begin{aligned} I(t) &= I(0) \exp(\gamma (R_0 - 1) t) \mathcal{F}(\mathrm{MGDD}(t)) \mathcal{G}((\mathrm{CDD}(t)) \\ &= I(0) \exp(\gamma (R_0 - 1) t) \Pi(t), \end{aligned} \tag{6}$$

where $\Pi(t) = \mathcal{F}(\mathrm{MGDD}(t)) \mathcal{G}(\mathrm{CDD}(t))$ is the cumulative probability of chronic infection dependence on temperature and $R_0$ bears the information on the vector density.

The spatial unit of the model is given by the resolution of the ERA5-Land data, for which we assume uniform conditions within each of the grid cells (approximately $9 \times 9\,\mathrm{km}^2$) in terms of vector population, susceptible vines and parameters that define the model. Risk outcome is calculated for each cell of the spatial raster individually; i.e., there is no simulated spread from one cell to another. Altogether, the equation representing the number of individuals with chronic infections in each cell $j$ at time $t$ is written as

$$I_j(t) = \underbrace{I_j(t-1)}_{\text{transmission layer}} e^{\gamma (R_0(j)-1)} \overbrace{\Pi_j(t)}^{\text{climatic layer}}, \tag{7}$$

where $I(t-1)$ is the number of chronic infections in the previous year ($t-1$) and $\Pi_j(t) = \mathcal{F}(\mathrm{MGDD}) \mathcal{G}(\mathrm{CDD})$ is the climatic layer that modulates the growth term and describes the cumulative probability of new infections becoming chronic in the time period between $t-1$ and $t$. The model assumes a homogeneous distribution of the vector population among the grid cells (same $\beta$ and then same $R_0(j) = R_0$) except for Europe, where information on the spatial distribution of P. spumarius is available (see "Methods"). In this latter case, a spatial dependent $R_0(j)$ is

incorporated into the model by considering the product of the homogeneous $R_0$ and the spatially-dependent climate suitability for vectors (Supplementary Note 2F).

To compute the epidemic-risk maps, we carried out a simple simulation summarised in three steps: (i) at the initial condition for the first year considered, $t_0$, each grid cell is seeded with a single infected plant, $I(t_0) = 1$; (ii) the simulation runs for a year and the incidence is calculated following Eq. (7); (iii) we seed again the cells for which the number of infected plants has vanished. In the last seven years of the simulation, there is no reseeding to allow the system to relax. This process is repeated until the final year $\mathcal{T}$. Finally, the risk index $r_j(\mathcal{T})$ is calculated from the final number of infected plants at grid cell $j$ as

$$r_j = \max \left\{ \frac{\log(I_j(\mathcal{T})/I_j(t_0))}{\gamma (R_0(j) - 1) \mathcal{T}}, -1 \right\}. \tag{8}$$

In this equation, $r_j$ implicitly delimits three differential risk zones in the maps: (1) non-risk zones where $r_j \leq -0.09$, and the number of infected plants decreases exponentially; (2) transition areas where $-0.09 < r_j \leq 0.075$, and (3) an epidemic risk-zone where $r_j > 0.075$ and PD can theoretically become established and produce an outbreak—the number of infected plants increases exponentially (see Supplementary Note 2E for further details).

Model performance was calibrated with observed records of PD presence in California and the southeast of the US, where the disease is well established. PD distribution data were collected from publications from 2001 to 2020. Publications were filtered by selecting only records where the pathogen detection on symptomatic grapevines was confirmed by PCR or Elisa. The exact coordinates of the records were taken when available in the publication or approximated to locality or county level to build the Supplementary Data 1[19,39,41,75–79]. For modelling purposes and to attempt a general rough estimate of the $R_0$ parameter valid for the entire US, we assumed a single vector with a uniform spatial distribution. We ran several model simulations with $R_0$ ranging from 1 to 14. Model prediction performance was estimated using a ROC curve by plotting the true-positive rate (TPR), calculated as the ratio (TP/TP+FN), against the false-positive rate (FPR), calculated as the ratio (FP/TN+FP), where PD absence/presence fulfil the following conditions: true positive (TP), PD is positive and $r > 0$; true negative (TN), PD is negative and $r < 0$; false positive (FP), PD absent but $r > 0$; and false negative (FN), PD positive and $r < 0$[80]. A different approach was followed to estimate $R_0$ for Europe given that PD is only present in Majorca and hence spatiotemporal data on the PD distribution is limited to the island. First, we estimated the transmission rate of the main European vector P. spumarius from the well-studied disease progress curve of the almond leaf scorch epidemic in Majorca. Then, using the known mortality rate of PD-infected vines $\gamma \sim 0.2\,\mathrm{y}^{-1}$ and the inferred transmission rate, $\beta = 0.8\,\mathrm{y}^{-1}$, the basic reproduction number for PD in Majorca yields $R_0 = \beta/\gamma \approx 4$. Finally, using data on the climate suitability of the vector in Majorca, $v = 0.8$, and inverting the relation $R_0(j) = R_0 v(j)$, we estimated $R_0 \approx 4/0.8 = 5$ as a maximal estimate baseline scenario for PD transmission in Europe (Supplementary Note 2D). This figure is not intended to be an exact estimate of $R_0$ but rather an average reference in our model in agreement with the lesser abundance of vectors relative to the US. Furthermore, since there is no information on the distribution of the potential vectors and no PD distribution data to calibrate, we also used a conservative $R_0 \approx 5$ scenario for the rest of the world.

**Distribution of wine-grape production areas.** Risk maps were focused solely on wine-grape regions excluding table and dried grapes-producing areas. Data on the vineyard surface in Europe were obtained from the CORINE land-cover map[81–83] (Fig. 6). The Nomenclature of Territorial Units for Statistics (NUTS) was used as a geocoding for the subdivisions of European countries for statistical purposes. To visualise the locations of the main growing regions in the risk maps, we included dots representing the distribution of the main wine-growing regions collected from official statistics and maps from the countries (Fig. 5).

***Philaenus spumarius* SDM.** The potential distribution of P. spumarius in Europe under current and future (i.e., 2050) climatic conditions was provided by Godefroid et al.[35]. Predictions were obtained using a generalised additive model and two bioclimatic descriptors i.e., a climatic moisture index for the coldest 8-month period of the year and the average maximum temperature in spring (March, April and May). Both descriptors reflect physiological constraints acting on life stages of the meadow spittlebug, particularly sensitive to spring temperature and humidity (eggs and nymphs), and were identified as good predictors of P. spumarius distribution[35]. We used the positive relationship between the climate suitability and spittlebug adult abundance[35] to assume no climatic constraints on vector population sizes at optimal climatic conditions ($v = 1$). Climatic suitability indexes, $v(x)$, were used to compute a spatially-dependent basic reproduction number, $R_0(x) = R_0 v(x)$. The linear dependence between the basic reproduction number and climatic suitability is justified by a vector-borne epidemic compartmental model (Supplementary Notes 2F and 4).

**Risk assessment by 2050.** Climatic variables were obtained with annual resolution by extrapolating the computed MGDD(t) and CDD(t) time series up to

2050. The observed trends of the time series were captured using a machine learning-based linear regression model while the interannual fluctuations were modelled by Gaussian noise (Supplementary Note 3). Future risk extrapolations were obtained as the average of $10^4$ simulations of this process. A correlative SDM was used to estimate vector spatial distribution in Europe using the global circulation model MIROC5 and greenhouse gas emission scenario RCP4.5, assuming moderate climate change[35]. Afterwards, the risk was computed following the same simulation procedure previously explained.

**Reporting summary**. Further information on research design is available in the Nature Portfolio Reporting Summary linked to this article.

## Data availability

The world hourly temperature data from 1981 to 2019 necessary to prepare Figs. 1 to 6 were taken from Copernicus ERA5-Land (namely the '2m temperature' field). The geographical vineyard distribution in Fig. 6 was taken from Copernicus Corine Land Cover database. Further source data for Fig. 1 is available in Supplementary Data 1. Points on the distribution of main vineyard zones given in Fig. 3 are available in Supplementary Data 3. Figure 6 is based on data in Supplementary Data 4. The source data of Table 1 is available in Supplementary Data 2.

## Code availability

We provide a library built in Julia to analyse the data outputs of ERA5-Land in GRIB format in this Github link. Furthermore, the simulation code and a small reproducible example are provided in this Github link.

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

## Acknowledgements

The authors thank the Balearic Islands Official Plant Health Laboratory (LOSVIB) and acknowledge funding from the Spanish Ministry of Science and Innovation through Grants RTI2018-095441-B-C22 (SuMaEco) and PID2021-123723OB-C22 (CYCLE) (A.G.R. and M.A.M.), and RTI2018-093732-B-C22 (PACSS) and PID2021-122256NB-C22 (APASOS) (J.J.R.) all funded by MCIN/AEI/10.13039/501100011033 and by ERDF "A way of making Europe", MDM-2017-0711 (A.G.R., J.G., J.J.R., and M.A.M.) funded by MCIN/AEI/10.13039/501100011033; The Ministry of Agriculture, Fishery and Food of the Government of the Balearic Islands under Grant (MEPRO 11876/2019). M.G. received the grant Ayudas a la Atracción de Talento Investigador (Ref: 2018-T2/ BIO-1137) funded by la Comunidad de Madrid.

## Author contributions

A.G.R., A.F., J.B., J.J.R., M.A.M., and E.M. planned and designed the research. M.M. and E.M. carried out the inoculation experiments. A.G.R., J.J.R., and M.A.M. conceptualised the mathematical model. A.F. and M.G. developed ecological models for the distribution of *Philaenus spumarius*. A.G.R., J.G., and J.B. built the risk maps. AGR and EM wrote the original draft. A.G.R., M.G., M.A.M., J.J.R., and E.M. reviewed and edited the final manuscript.

## Competing interests

The authors declare no competing interests.
