## [Peer Review File · Communications Biology]

Reviewers' comments:

Reviewer #1 (Remarks to the Author):

Please see attached file

Reviewer #2 (Remarks to the Author):

This study attempts to predict how the risk of Pierce's Disease (PD) in grapevine varies spatially on a global level, with a particular focus on grapevine production areas. The novelty of this study compared to previous work is the use of a mechanistic model informed by the transmission dynamics of the pathogen rather than a purely correlative approach. This is a novel approach for this important pathogen, and may increase the robustness of the results and allow better investigation of important factors influencing pathogen distribution.

The study brings together a wide range of data - ranging from experimental data at the individual plant level up to satellite data at the global level - and with this, a range of analysis approaches. For this, the authors are to be commended. However, I found that clear explanation of the approaches used is sometimes lacking in both the main MS and in the SI. One major suggestion is therefore that the authors better describe their methods. However, this also means that there is a risk that I have misunderstood some aspects of the study, for which I apologise to the authors in advance.

My main concerns about the study are as follows:

1. The authors argue that their experimental data provides valuable information on the temperature requirements for development of PD (which is captured in their MGDD parameter). However, as far as I can see, the authors are just monitoring the development of symptoms following inoculation with *X. fastidiosa*. I accept that the temperature needs to be appropriate for the pathogen to multiply and these symptoms to develop, but I have trouble seeing how the authors can argue that this development of symptoms directly reflects the influence of MGDD (that is, it is plausible that very similar patterns could potentially be observed with very different accumulations of MGDD). I appreciate that the author's approach is also supported by in vitro evidence of the influence of MGDD, and I suspect that the authors' approach is a reasonable one, but I feel that a little caution in the interpretation of MGDD as the cause of the disease progression (and a little discussion of how this uncertainty would impact upon the conclusions) would be useful, given the influence this could have on the model predictions.
2. I have a lot of trouble understanding how the authors are capturing CDD, as the paragraph explaining this in the SI jumps back and forth between different ideas. As far as I can see, it appears that the authors are arguing for the use of the average minimum temperature of the coldest month as a proxy for CDD, since both are correlated. However, I may have misunderstood this and feel that a better explanation is required here. It is also not explained how the function used for CDD is calculated, and upon which data this is based.
3. The discussion of R_0 estimation is also a little unclear. As I see it, the authors use at least two different approaches to estimate R_0 (for ALS in Europe, PD in the USA, and potential PD in Europe), but these are minimally explained. Presumably the method based on the models (used for the European estimates for ALS and PD) involves the analysis of next generation matrices, but if so the method should be briefly explained. It may also be useful for the authors to point out to readers that they are not explicitly modelling vector infection, as the R_0 calculations (and likely the estimates themselves) would be different if so.

4. I also have some concerns about the assumption that the R_0 is either fixed or scales linearly with vector climate suitability. Here, the authors are capturing a lot of complex processes (including vector feeding behaviour and movement) in a very simple way. Although I do appreciate that such simplifications are needed in models such as this (and again, I do think that it is a justifiable assumption for the authors to make), I think that a brief mention/discussion of this assumption would be useful.

Some other minor points are listed below, indexed by page number:

Page 1

- "> 560 plants species" should be "> 560 plant species"

Page 2

- It would be useful to clarify the meaning of "incidence" as it varies between disciplines, or use another term

- I don't understand the authors' points about "early infections" - can this be clarified please?

- "none of these works has explicitly included" should be "none of these works have explicitly included"

- I am not sure what the authors mean by "invasive criterion" in the context used here. Can this be clarified please?

- The wording "By reducing the global risk of PD" is misleading (since the true global risk of PD is unchanged). Something like "By estimating a lower global risk of PD" would be more correct.

- "chronic infections" is not defined when it is first introduced. Could the authors define what they mean by this please?

Page 3

- The caption for Figure 1B does not explain what the lines represent (presumably they are means). Could this be stated please?

Page 5

- It is a little unclear to me what "full presence of the vector" really means. Does it mean that vector densities would be expected to be maximal? Or that there are no climatic constraints on vector population sizes? I think that this should be explained as this is the estimate being used to scale the R_0 value.

Page 7

- The paragraph starting "knowledge" starts in lower case. I assume that this is just a typo rather than words being missing.

- I think that the statement that there is "enough information on the principal vector" in Europe is debatable (this word could just be removed). Also, I suspect that this should be citing Figure S10 rather than S7.

- "indirect empirical evidence of this non-linear relationship" is better phrased as "indirect empirical evidence of a non-linear relationship"

Page 8

- "our risk maps go further beyond by incorporating" is better phrased as "our risk maps go further by incorporating"

- I think that the statement "a comparison of PD risk maps for Europe with different R_0 suggests for non-Mediterranean areas the need to stress more surveillance on the introduction of alien vectors rather than in the pathogen itself" requires more explanation.

Page 9

- "biweekly" is unclear as it can mean twice weekly or every other week. Could the authors clarify which of these is meant?

- "The mathematical relation bacterial population growth" should be "The mathematical relationship between bacterial population growth"

- Unless I have just missed it, I cannot see where the authors describe how CDD is captured in this Methods section. Can this be included please?

Page 10

- The gamma in equation 1 is not defined until later in the text - can the authors define it here please?

- I think that the authors should point out after "a spatial dependent $R_0(j)$ was incorporated" exactly how this was done (i.e. the product of R_0 and spatially-dependent climate suitability for vectors).
- I think that "at each time" would be better phrased as "at each timestep" to point out that a discrete time model was used here
- "P. spumarius distribution [35] and" is missing an left parenthesis before the reference.

Page 12

- The site <http://pdrisk.ifisc.uib-csic.es/> does not appear to be accessible to the public. Can this be corrected before publication please?

Throughout the MS

- I also note that SI sections are sometimes referenced explicitly and sometimes as "SI Appendix". Could one or the other be used consistently please?

SI

Page 7

- The sentence "and thereby the bacterial population after a given time t is related to the MGDD by Eq. (S9)" would be better placed directly after S9 than in its current position.

Page 8

- The term "continuous factor" is confusing to me, as factors are commonly considered as categorical. I think that "function" would make more sense. I also think that it would be useful to state that this function is bounded by 0 and 1. "Factors" are mentioned elsewhere in the text, and I think that these mentions should also be changed to "function".
- I think that the whole CDD paragraph needs to be rewritten/restructured (and the parameters explained) as I currently do not understand it.
- The statement "With this choice, the 75% of the infected vineyard population get recovered in the winter curing threshold" should be rewritten as something like "With this choice, 75% of the infected vineyard population recover at the winter curing threshold"

Page 9

- The authors should make clear whether the R_0 of 8 estimated for the USA is for ALS or PD.
- I think that more explanation of the climate suitability for the vector estimates is needed - just a quick summary of how they were calculated and what exactly they mean.
- I do not understand why Equation S16 needs to be converted into a map in order to account for MGDD and CDD changing over time - could the authors clarify this please?
- In the footnote at the bottom of the page, Spumarius is capitalised when it shouldn't be.

Page 10

- The authors state that the upper limit of the Transition-risk zone is $r_{\{\max\ trans\}} = 0.075$, but the lower limit of the Epidemic-risk zone is 0.1. Should this instead be $r_{\{\max\ trans\}}$?

Page 11

- There are two question marks in the text.

Page 17

- There is a plus and a minus together in the equation of plot B. Although strictly speaking this is not incorrect, for consistency with plot A I think the plus can be removed. I also don't see the particular value of the intercept in either of these plots - it would be more useful and informative to set it to a useful date (for example, 2019) rather than year 0.
- It is very unclear to me what "the areas encompassed below the CDD i 314 line" in Figure S6 means. Could the authors clarify please?

Page 18

- More information is needed in the legend of Figure S7 - such as the source of the data, the host, and so on. Can the authors add this in please?

Pages 34-35

- Can the authors please sort the formatting of the titles of references 3, 4, 10 and 17 and the DOI and citation for reference 17?

Reply to Reviewers Comments

We include the reviewers' comments, point by point, in blue and then develop our answer (in black)

Reviewer 1

I had a pleasure to be assigned to give my opinion about the manuscript which describes the modelling work with regards to the potential risk of establishment of Pierce's disease as an endemic disease. The report itself is well written and the amount of the work is commendable. However, there are several concerns which should be addressed by the authors before consideration for publication.

The main point of concern is the presentation of field data, analysis, and subsequently the development and parameterization of the epidemiological model. Furthermore, this as another epidemiological study is built on several assumptions, from the choice of modeling structure to their parameterization. These choices need to be transparently discussed.

In terms of the structure, methods section in the main document needs to contain enough information for potential reader to understand reported results. Discussion needs work to reflect the some of the points mentioned above and improvement of the structure.

Some comments are provided below. Apologies for somewhat hectic organization arisen due to several concerns occasional need to go back and forward to understand the work. I am sure that if some of these suggestions were to be implemented manuscript would be more accessible and perhaps, I would have easier time reading it second time around.

We appreciate the reviewer's general comments. We have worked along the lines mentioned by the reviewer, and have improved the presentation of field data, analysis and the discussion of the parameterization and formulation of the epidemiological model. In the original manuscript, the results were presented before properly discussing the model, and this has been addressed in the revised manuscript.

General suggestions:

Please italicize Latin names throughout.

This has been addressed in the revised manuscript.

Please check for double spaces (seen a few, e.g. Pg5col2par2: Much less surface.)

We acknowledge the reviewer for pointing out this typo. This has been amended in the revised manuscript.

I would strongly suggest including table of the many abbreviations used throughout.

We agree with the reviewer. Table I (in the main text) has been created.

A small reproducible example wherever possible, especially in terms of model development should be provided. Authors have used freely available data and computing software and in the same spirit of scientific development, it would be beneficial to provide these resources to contribute to the development of the field. Additionally, it would make the review process easier.

Thank you for this suggestion. Indeed, a GitHub repository with the main code used for the work was already public and cited in the manuscript. This repository consists of a self-made library used to compute MGDD and CDD from the ERA5-Land dataset, which is the most technical and difficult part of the work.

Now, we have created a new repository with a small reproducible example showing the whole workflow and code to simulate a single year of PD progression (we only provide the example for a single year because each year consists of ~5GB of data).

We provide a script to download the necessary data files. The script uses a self-made library to compute MGDD and CDD from ERA5-Land files and a jupyter notebook file to run the simulation itself. All requirements are specified in the repository publicly accessible at: <https://github.com/agimenezromero/Global-risk-predictions-for-Pierce-s-disease-of-grapevines>

More specifics suggestions and comments:

Authors need to define what kind of risk are they refereeing to (the tile and in general). Also, in the abstract, what kind of decision-making? E.g. there needs to be clarity if this tool and/or platform are to be used to inform policy makers, or in grapevine IPM, solely for predictions disease establishment of PD as an endemic disease?

For the sake of brevity, we used a condensed title. Our work addresses the risk of establishment and predicts the epidemic growth rate potential (see in the Modelling disease section, the risk index dual interpretation). Accordingly, we have updated to the more suitable title “Global predictions for the risk of establishment of Pierce's disease of grapevines”.

The abstract is limited in the number of words. Still, we have specified that the decision-making refers to plant health. Our aim in this work is to study the risk of establishment of PD. While the model could be extended to assess the efficiency of different scenarios with and without PD control policies, this is beyond the scope of the present work. Indeed, it will be certainly an interesting future project. We have modified the discussion section to leave this point clearer. On the other hand, we believe that the webpage created in this work is an important tool to visualize where there is a risk of establishment of PD and where the disease incidence could be significant. This tool should be useful either for policymakers, grapevine IPM, particular farmers or in general for anyone who is interested in understanding the factors leading to PD.

Pg2col1par4 Suggest: weather-driven models. Climate is a considered to summary of weather over a long period of time.

The reviewer is completely right! Climate refers to averages of weather over long periods of time. The temperature hourly forcing that we have used in the model is weather forcing. However, the establishment of PD is the result of a longer

process. It includes the effect of MGDD and of winter curing over a long period. It is in this sense that we talk about climatic suitability. In some parts of the text, we have been more explicit in terms of using temperature-forcing instead of climate, such as in the abstract in which we have changed it to “temperature-driven dynamic model”.

Results

First section of results should include more results rather than the repetition of methods.

The reviewer is right! This was partly due to placing the model in the methods section. We now include only the results of the inoculations and have deleted methods explanations in the first paragraph of the first section. We also have moved part of the results of the inoculations from the Supplementary Information to the Results section and left only some of the method explanations in the second paragraph of the Result section when it is needed to follow the argument. Aspects of the model that are discussed in Results are no longer repeated in Methods in the revised manuscript.

Pg2col2par3 I am unfamiliar with the term field-temperature conditions. Please simplify terminology.

Indeed, this was not a good expression. “field-temperature conditions” has been changed in the revised manuscript to “air temperature” when referring to temperature and “field condition” when referring to conditions similar to those found in the vineyards.

The authors correctly point out that the use of means contributes to loss of accuracy by ignoring the variance. Hence, their use of means in model development and possible implications and oversimplifications on the results of this study should be addressed and discussed.

In the case of the temperature, we are not using daily means but hourly values to capture these fluctuations (variance) in the model. This is important especially in the case of MGDD and CDD, because they are nonlinear functions. The model is now better exposed in the Results section and these questions, including the assumptions, are clearly explained.

I have several concerns about the development of the epidemiological model. All of the graphs with points representing means should have error bars to show the variation within the group.

We apologise because the origin of the dots was not well explained in figure 1C. They refer to the proportion of infected plants at different MGDD levels. We now have included the 95% confidence interval bar at the MGDD levels. We have not included the variation within the groups in the graph because it looks quite fuzzy (see **figure 1** below) and this information is provided in Table S1. On the other hand, the data from Fig 1A is taken from a published article (Ref. [38], Feil & Purcell, 2001) and it does not have error bars.

Figure 1. Relation of the probability of chronic infection and the climatic functions MGDD and CDD. Different colours in the symbols correspond to the diverse scion/rootstock combinations.

It is certainly true that the use of growing degree days as a binary values does not represent the realistic representation of biology of the pathogen life cycle. I am however surprised that authors used the approach described in supplementary material S1. this could have been done in a simpler manner using beta function often applied for such purpose^{1,2}

We apologize but we are not sure what the reviewer refers to as binary values in Supplementary material S1. In the survival analysis, we do use binary values to censor the event of interest which can be represented by different families of functions including a beta logistic model. On the other hand, the beta function described in the references provided by the reviewer is an interpolating, smooth function that passes through three points: the minimum, maximum and optimum temperatures. It is customarily used as a smooth description when only information about the cardinal temperatures is available. However, in the study of Feil & Purcell (2001) [38] they provide information about the growth rate at many temperatures. Their results fully agree with a fundamental description of the effects of temperature in chemical and biological systems derived from Arrhenius' equation. Based on that work, we obtain a useful linear approximation for the temperature range relevant for our study (S2.A) and similar to Degree Days descriptions, which we call MGDD. Moreover, the excellent agreement of the MGDD representation with the available experimental data of [38] is shown in Fig. S2. In contrast, as a further check, we have found a very poor agreement between the over-imposed predictions of a beta function on the experimental data of Feil & Purcell (2001) [38].

I remain unclear about the section C in the relation between MGDD and disease expression. I am unclear about the figure which says but there were 36 varieties

included and yet in methods there's a number of 36 and 54? Besides, I can only see 16 points on figure SD? It would be very good to see points from different years in different colors.

The reviewer is right, there were several mistakes (e.g., 54 vs. 57) in the explanation. In the new manuscript version, we clearly explain in the Results the relation between MGDD and disease expression and more details are provided in the Methods. This section C of the Supplementary information has been removed since it is no longer needed.

Are there any photos of the experimental setup? It would help us visualize the experiment. Another major concern is the placement of the weather station and sensors. This needs to be described in detail.

In the Methods, we briefly describe that "Plants were randomly distributed in rows of 12 plants along an insect-proof net tunnel exposed to air temperature" and then say that "Mean hourly temperature data were recorded during the 3-year trial with an automated weather station located outside the insect-proof net tunnel. We have also included a photo in the Supplementary information (Fig S2) showing the size and volume of the greenhouse (an insect-proof net structure) in relation to the plants and an example of the arrangement.

It appears that the trial was not replicated across seasons but that certain varieties were tested in different seasons. This must be clear in the main text. With a full appreciation of time and effort which go into these type of experiments on grape wine, I find it difficult to ascertain the thought of the models developed on a single site to be representative in any other climate without further validation in other climate, pathogen races and local varieties. However, it is important to note here that, comparing to many other experimental in vitro studies, in vivo approach is certainly more representative of natural epidemic development.

The focus of the article has been to use the inoculation data and the relationship between temperature and symptom development to build an epidemiological model and transform this information into risk maps. The information on the inoculations could constitute an article on its own, so we have relegated most of the information to the Supplementary material. All the information is available in the datasets and tables for any researcher interested in using it.

Our aim is to model the general average behaviour of the worldwide population of *Vitis vinifera* varieties (>1000 varieties) to the development of PD symptoms in relation to the accumulated temperature. We have used a sample of 36 varieties and 57 scion-rootstock combinations, which represent more than the 75% of the worldwide surface of vineyards in wine-producing areas. This point is now further stressed in the manuscript. Nonetheless, the question of whether models developed on a single site are representative for extrapolating to other climates is a reasonable one. The reviewer 2 also questioned it in a similar manner.

It would be ideal to have other inoculation experiments with the same Xf strains in other continents with other climates and similar facilities, and replicated them in two or three years. For obvious reasons (e.g. quarantine pathogen, enormous facilities and economic resources) this has not been possible. However, we have now a four-year experience in inoculating grapevines with XfPD. If we have a look

at the variety Tempranillo, for example, we have inoculated during the three years, 12 Tempranillo-rootstock combinations. We can observe significant differences among the Tempranillo-rootstock combinations, years and even between the virulence of Xf strains when analysed with a GLM (see Supplementary Data 1 and Table S1). Our experience indicates that the results of the inoculation and symptoms development show some variation due to the inoculation process, the erratic movement of the bacteria within the xylem vessels and the virulence lost by conservation of strains at -80°C. Rootstock has also a significant effect.

The main point is whether similar symptom development patterns could be produced with different MGDD values and this could invalidate our MGDD estimations. Given the impossibility of doing inoculations in different continents and climates, the closest approach is to verify how in the field the appearance of PD symptoms correlates with the accumulation of MGDD in other areas where PD is established. Thus, the wine-producing areas of the Napa and Sonoma Valleys have a Mediterranean climate like the Balearic Islands but are somewhat cooler. Unlike the Balearic Islands, the hot summer months extend into September and part of October in California and Oregon, while the months of June and July are clearly cooler than in the Mediterranean. Consistently, PD appears in late summer in the Sonoma and Napa valleys and is barely observed in years with relatively cool summers, whereas PD symptoms appear at the very end of July in Mallorca. In southern Oregon, the accumulation of MGDD is not sufficient according to our model and therefore no PD is observed. On the attached web page <http://pdrisk.ifisc.uib-csic.es/> it can be done the simulations and verify how our MGDD approximations make good forecasts for California and Oregon. It is also noteworthy that insect vector *Homalodisca vitripennis* was detected for the first time in Willamette Valley Oregon in 2000. Despite using a $R_0 = 16$ the climatic conditions for Oregon, the temperature conditions are not conducive to PD, as it accurately shows our model.

Figure 2. Comparison of MGDD in Mallorca, northern California and Oregon. MGDD values above 1000 are approximately reached one month later in California than in Mallorca which shows a good correspondence with the lag of symptoms in the field. When included the probability of chronic infection (right) the differences widen. In Oregon, the climatic layer is not enough to produce PD.

To further show uncertainties related to the variation of MGDD among grapevine varieties (also reclaimed by reviewer 1), We now have included in Fig 1C the 95% confidence interval bar at the MGDD levels. We have also included in the Discussion the following sentence:

“The physiological basis of the plant-Xf interaction leading to symptoms development is poorly understood. Although the temperature is indeed a fundamental and dominant environmental factor, others such as drought, nutrient status or crop management may modulate symptom expression and hence add an error in the MGDD parameter not measured in this work. Nonetheless, we deem the error range would be smaller than the differences of MGDD among varieties (i.e., regional differences) and smaller than the interannual MGDD oscillations in most locations”

What are the characteristics of the climate change scenario used? While this is not main area of research, I am aware that these scenarios do have certain characteristics.

This is, indeed, a very good point! Ideally, it would have been good to perform one of such predictions, namely with the +2 °C, +3 °C, etc. climate change scenarios. These calculations would involve implementing the calculation of the MGDD and CDDs with these scenarios averaged over several models, which is far from trivial. We have not used any proper climate change scenario. We extrapolated instead a known time series of MGDD and CDD calculated with the available historic data using a ML linear model (Sklearn LinearRegression package). Stochastic fluctuations (Gaussian noise) in the historic MGDD and CDD series were also included in order to reflect the inter-annual variability in the machine learning linear extrapolation. This question is well explained in the new Methods section.

Our goal is to present a first rough estimate of the areas that present more risk due to global change effects while acknowledging that the predictions are not expected to be accurate in some cases. This is because both the MGDD and CDD functions are non-linear. Thus, if one looks at the MGDDs, while in some areas an increase in temperatures may lead to an increase in MGDD values, in others it may lead to increased exploration of temperature ranges above the optimal temperature, with a smaller contribution to MGDD, or even zero if they are above the maximum temperature. Consequently, the linear prediction may overestimate the risk, which could actually decrease due to global change effects. A proper study on the effects of global change on the risk of establishment of PD using the present model with different global change scenarios would certainly be a goal to be carried out in the future. This point has been further explained in the Discussion section of the manuscript.

Could the initial number of plants have some spatial structure? I am not sure that using a single plant per pixel is satisfactory. Could have authors made more spatially explicit model with randomized points of entry (assign “diseased” some pixels of the matrix) and monitor the spread?

This is a good point, but our model does not account for the spatial structure for disease spread. Our model unit (cell) is provided by the resolution of ERA5-Land data in pixels of $0.1^\circ \times 0.1^\circ$ (around 9 km x 9 km). In no case we intend to model the two-dimension disease spread among the cells, as has been done in other works (Schneider et al. 2020; White). Each pixel is thus an independent unit. We assume that every pixel can be described using a compartmental model (a modified SIR model) and, at the same time, we assume that the number of infected hosts is small compared to the population of susceptible hosts. Thus, one can write Eq. (1).

The growth rate of the infected population does not depend on the initial conditions in Eq. (1). In fact, in Eq. (2) the risk is calculated from the ratio between the infected population at time τ and the initial population. This ratio is independent of the initial seed in the pixel because the exponential does not depend on the number of infected hosts. Therefore, the use of one initial infected plant is suitable. We are only interested in the initial phase as we are only modelling the risk considering whether the infected population will grow or not after the introduction of a hypothetical infected plant. In practical terms, these assumptions imply that there are enough vectors and susceptible hosts to spread the disease if the temperature conditions are adequate. This point is better explained in the new version of the manuscript.

In terms of structure, while I would commend the attention to detail as authors provide detailed description of modeling process in supplementary materials, I did not find the description of methods in the main text to be sufficient as a standalone. While this might not be a barrier for a common reader as it is for a reviewer, this would require some attention.

We agree with both reviewers and the Editor, and, accordingly, we have provided a description of the model in the main text. Since the model is introduced in this work and is one of our main results, it is now explained in the Results section, leaving only a few technical details in the Methods section. We believe the new structure reads better and avoids repetition of methods in the results and the need to go back and forward to understand the work.

The opening statement in the discussion section should put the results of the study into perspective rather than being focused on the methodology. Big part of the first few paragraphs sounds like something that should be a part of a section named conceptual framework, which is normally between study goals and methods.

In the revised version of the manuscript, the model description is included as part of the Results. The model is new and the process to building and calibrating it is not trivial. We have introduced a few paragraphs for explaining these new developments, but this is not a recap of methodology. We believe that the first paragraph of the revised Discussion now makes more sense. We open the discussion by indicating what is new about our study compared to what has been done previously and how the incorporation of epidemiological models in the construction of risk maps is an approach with high predictive power. We also hope that in the new version of the manuscript this is clearer.

I tried accessing the web page of the prediction tool but used authentication is required for which credentials are not provided.

We apologize there has been a technical problem with the address. The web page <http://pdrisk.ifisc.uib-csic.es> is now public and no credentials are needed to access it.

Pg7col1par2 I would suggest using the phrase disease development instead of symptom development. Symptomatic stage here's the last stage of the disease development and as such should be used appropriately. This should be consolidated throughout.

The suggestion of the reviewer is correct; the symptomatic stage is a consequence of disease development. However, we use 'symptom development' in many passages of the manuscript to stress that we have only monitored the symptoms over time and not, for example, the bacterial concentration *in planta*. It is a nuanced matter, but, strictly speaking, we have not addressed the asymptomatic stage of the disease. In other cases, we take into account that PD is a pluriannual progressive disease, so every year there is a cycle of symptom development.

In the place suggested by the reviewer, we agree it is more appropriate and we have changed it for "PD development can be characterized as a continuous thermal-dependent process within the cardinal temperatures of X_{fPD} growth [53]."

Pg7col2par2 Capital letter. + correct the last sentence

Corrected.

Pg7col2par2 The use of language and clarity should be corrected throughout. The second sentence here, while being somewhat unclear, is correct.

For clarity, we have modified the sentence as "*Knowledge of the insect distribution is crucial for predicting epidemic outbreaks of endemic diseases, as well as the risk of invasion by emerging vector-borne pathogens ([56, 62], (cf. [49])). Given the great diversity of known and potential vector species that can transmit PD [30], it has not been possible to include in the model each one of the particular vectors of a region. Therefore, when evaluating the risk of PD on a global scale, we have considered a homogeneous spatial distribution of the vector (a fixed value of R_0 in all the pixels), except in Europe where there is information on the main vector (Figure S7). As expected, the European case shows how models that assume a homogeneous spatial distribution of the vector generally produce epidemic risk zones with higher risk indices than models that include a heterogeneous spatial distribution (Table S2 vs. Table S7). This lack of vector information is one of the main reasons why vector-borne plant diseases are overestimated in risk assessments*".

#####

Reviewer #2

This study attempts to predict how the risk of Pierce's Disease (PD) in grapevine varies spatially on a global level, with a particular focus on grapevine production

areas. The novelty of this study compared to previous work is the use of a mechanistic model informed by the transmission dynamics of the pathogen rather than a purely correlative approach. This is a novel approach for this important pathogen, and may increase the robustness of the results and allow better investigation of important factors influencing pathogen distribution.

We thank the referee for this comment.

The study brings together a wide range of data - ranging from experimental data at the individual plant level up to satellite data at the global level - and with this, a range of analysis approaches. For this, the authors are to be commended. However, I found that clear explanation of the approaches used is sometimes lacking in both the main MS and in the SI. One major suggestion is therefore that the authors better describe their methods. However, this also means that there is a risk that I have misunderstood some aspects of the study, for which I apologise to the authors in advance.

We acknowledge the reviewer for this constructive remark. Actually, the model is an integral part of the Results of this work, and, so, following also the remarks of Referee #1, we now present the model in the Results section, before exposing the risk predictions. Further detailed aspects and technical discussions can be found in the Methods section and Supplementary Information. We hope that this order will allow a better understanding of the model and its predictions.

My main concerns about the study are as follows:

1. The authors argue that their experimental data provides valuable information on the temperature requirements for development of PD (which is captured in their MGDD parameter). However, as far as I can see, the authors are just monitoring the development of symptoms following inoculation with *X. fastidiosa*. I accept that the temperature needs to be appropriate for the pathogen to multiply and these symptoms to develop, but I have trouble seeing how the authors can argue that this development of symptoms directly reflects the influence of MGDD (that is, it is plausible that very similar patterns could potentially be observed with very different accumulations of MGDD). I appreciate that the author's approach is also supported by in vitro evidence of the influence of MGDD, and I suspect that the authors' approach is a reasonable one, but I feel that a little caution in the interpretation of MGDD as the cause of the disease progression (and a little discussion of how this uncertainty would impact upon the conclusions) would be useful, given the influence this could have on the model predictions.

This is indeed a very good point! Considering the temperature as the sole climatic influence that affects the development of Pierce's Disease, and neglecting other effects, like humidity and others is an approximation. Ideally, one would perform similar inoculation experiments like those described in the manuscript in different wine-producing regions in the world with different climate patterns to validate the predictions. It will be certainly interesting if such inoculation experiments would be carried out in the future. Given the impossibility of doing inoculations in different continents and climates, the closest approach is to verify how in the field the appearance of PD symptoms correlates with the accumulation of MGDD in other areas where PD is established. Thus, we have compared the predictions of the

model in some spots of the Mediterranean basin with wine-producing locations in California and Oregon, finding that the model is quite predictive. Focusing on the MGDD in a single summer season, we observe a good correspondence between the MGDD in the inoculation experiments and PD symptom development in vineyards of Majorca in late July (cf. Fig 1a). This area is characterized by relatively hot and dry months in June and July, during which MGDD grows relatively fast until the value that yields PD with high probability. In contrast, the wine-producing areas of Napa and Sonoma valleys in California and also wine-producing areas in Oregon are characterized by a different Mediterranean climatic pattern. Namely, there is less heat accumulation at the beginning of the summer due to local fogs. Indeed, PD symptoms manifest in the late summer in Napa and Sonoma, and it is not observed in cool summers. In Southern Oregon the MGDD does not reach the threshold value for PD, and, in fact, PD is not observed (**Figure 2**). Although we agree that this is not a demonstration, we find that MGDD consistently predicts the appearance of symptoms of PD in these two areas. This can now be verified on our web page [57]. Finally, in 2000 the vector *Homolodisca vitripennis* was detected in Willamette Valley (Oregon), and this did not lead to a PD outbreak. Our model (with $R_0=8$) correctly predicts that the MGDD in this area does not lead to PD establishment.

Figure 2. Comparison of MGDD in Mallorca, northern California and Oregon. MGDD values above 1000 are approximately reached one month later in California than in Mallorca which shows a good correspondence with the lag of symptoms in the field. When included the probability of chronic infection (right) the differences widen. In Oregon, the climatic layer is not enough to produce PD.

We have also included in the Discussion the following sentence to reflect the uncertainties:

“The physiological basis of the plant-Xf interaction leading to symptoms development is poorly understood. Although the temperature is indeed a fundamental and dominant environmental factor, others such as drought, nutrient status or crop management may modulate symptom expression and hence add an error in the MGDD parameter not measured in this work. Nonetheless, we deem the error range would be smaller than the differences in MGDD among

varieties (i.e., regional differences) and similar to the interannual MGDD oscillations in most locations”

2. I have a lot of trouble understanding how the authors are capturing CDD, as the paragraph explaining this in the SI jumps back and forth between different ideas. As far as I can see, it appears that the authors are arguing for the use of the average minimum temperature of the coldest month as a proxy for CDD, since both are correlated. However, I may have misunderstood this and feel that a better explanation is required here. It is also not explained how the function used for CDD is calculated, and upon which data this is based.

We recognize that the CDD part of the model describing the effect of winter curing was not as detailed and comprehensive as the MGDD in the submitted version of the manuscript.

Regarding CDD, the available information in the literature (Lieth et al.) is not enough to build a function capturing the probability of recovery. We had thus to rely on the CDD measures, the standard degree-days accumulated below the base temperature of 6°C suggested in [6]. Additionally, we have checked whether this CDD metric is compatible with the main observation about winter curing of Purcell [40]: PD is not found in zones where the minimum average temperature of the coldest month is below -1.1°C. On the other hand, we have calibrated the CDD function in such a way that at the CDD threshold 75% of infected plants are cured, and that the combined effect of both the F(MGDD) and G(CDD) probability functions yields results that are compatible with the observed distribution of PD in the southeast of the United States.

Accordingly, we have expanded the explanation in the revised manuscript: “We then investigated the probability of disease recovery by exposure to cold temperatures. However, unlike the approach followed to relate MGDDs to $X_{f_{PD}}$ growth and symptom development, there are sparse experimental data to lean on. Following the work of Lieth et al. and an inverse analogy with the MGDDs, we assume that cold temperatures contribute to the decrease of the bacterial population in planta. Thus, the accumulation of cold degree-days (CDD) is expected to be a proper metric to describe the probability of recovery of infected plants. To account for the effect of winter curing, we related the accumulation of cold degree-days (CDD with base temperature = 6 C, i.e., $CDD(t) = \sum_i 6 - T_i$ for $T \leq 6$ C) with the average minimum temperature of the coldest month, T_{min} , isolines in the United States proposed by Purcell (available in [41]) as reference zones where PD is rare (T_{min} between -1.1 C and 1.7 C), occasional (1.7- 4.5 C) and severe (> 4.5 C) (Fig. 1B, Methods). This allows to relate accumulated CDD to the probability of disease recovery) using a generalised sigmoidal function, G(CDD) (see blue curve in Fig. 1C and Methods).”

3. The discussion of R_0 estimation is also a little unclear. As I see it, the authors use at least two different approaches to estimate R_0 (for ALS in Europe, PD in the USA, and potential PD in Europe), but these are minimally explained. Presumably the method based on the models (used for the European estimates for ALS and PD) involves the analysis of next generation matrices, but if so the method should be briefly explained.

We admit this part needed more explanations in the main text. The analytical derivation of R_0 can be consulted in the Supplementary information. Although R_0 could be calculated with next-generation matrices, we did it directly from the linear stability analysis of the fixed point, as it is easier for the SIR model.

All the R_0 values used throughout the manuscript are for PD; the only difference is how they were estimated. In the USA, given that there are available records of PD distribution, a ROC curve was computed to assess the model accuracy under different R_0 scenarios, calibrating the model. This approach could not be followed for Europe due to the lack of data. An alternative estimation taking as basis ALSD was performed. This is now explained in the Results in the model calibration and validation section:

“Model calibration and validation. *To attempt a rough estimate of the R_0 parameter in the United States assuming a uniform spatial distribution of the vectors, we ran several model simulations validating the spatiotemporal distribution of PD from data collected from publications between 2001 and 2015. We found $R_0 = 8$ as the optimal parameter for maximising the area under a ROC curve (Supplementary Fig. S4). In general, our model returns an accuracy of more than 80%, except for 2006, due to data retrieved from a study in the Piedmont region in the USA at the limit of the transient-risk zone (Supplementary Fig. S6 and Table III). A different approach was followed to estimate R_0 for Europe given that PD is only present in Majorca and hence spatiotemporal data on the PD distribution is not available. First, we inferred the transmission rate, $\beta = 0.8$ years⁻¹, of the main European vector, *P. spumarius*, from the well-studied disease progress curve of the almond leaf scorch epidemic in Majorca and the estimated almond tree mortality rate, $\gamma \sim 1/14$ years⁻¹. Then, using the known mortality rate of PD-infected vines $\gamma \sim 1/5$ years⁻¹ and the inferred transmission rate, we could determine $R_0 = \beta/\gamma = 4$ for PD in Majorca. Finally, using data on the climate suitability of the vector in Majorca, $v = 0.8$, and inverting the relation $R_0(j) = R_0 \cdot v(j)$, we estimated a lower $R_0 = 4/0.8 = 5$ as a baseline scenario for PD transmission in Europe (Supplementary Section S2 D). This figure is not intended to be an exact estimate of R_0 but rather an average reference in our model in agreement with the lesser abundance of vectors relative to the United States. Furthermore, since there is no information on the distribution of the potential vectors and no PD distribution data to calibrate, we also used a conservative $R_0=5$ scenario for the rest of the world.”*

It may also be useful for the authors to point out to readers that they are not explicitly modelling vector infection, as the R_0 calculations (and likely the estimates themselves) would be different if so.

We acknowledge that our SIR approach does not explicitly account for the vector. However, if the decay-population time scales of hosts and vectors are very different it can be reduced to a SIR model. This is explained now in S4 in the SI and recently shown in a publication [arXiv:2202.05598](https://arxiv.org/abs/2202.05598) by two of the authors for a quite general vector disease model with host-vector interactions. This is the case of PD in which the lifetime of the vector and the time span of the disease progress on the plant are very different, which gives ground to using the SIR model in our study.

4. I also have some concerns about the assumption that the R_0 is either fixed or scales linearly with vector climate suitability. Here, the authors are capturing a lot of complex processes (including vector feeding behaviour and movement) in a very simple way. Although I do appreciate that such simplifications are needed in models such as this (and again, I do think that it is a justifiable assumption for the authors to make), I think that a brief mention/discussion of this assumption would be useful.

As pointed out by the referee, the linear scaling of R_0 with vector climate suitability needs to be clarified. In several compartmental models of vector-borne diseases, the basic reproduction number (R_0) scales linearly with the vector population, which justifies our assumption. In Section 4 of the Supplementary Information, we have included an analytical derivation of the linear dependence between R_0 and the vector population (i.e. the number of vectors). Then, assuming that climatic suitability (i.e. probability of presence) is directly related to the number of vectors we obtain the linear scaling between R_0 and climatic suitability for vectors.

Some other minor points are listed below, indexed by page number:

We acknowledge the referee for his/her detailed reading of the manuscript and for pointing out these typos and other minor comments, that certainly improve the manuscript.

Page 1

- "> 560 plants species" should be "> 560 plant species"

This typo has been corrected.

Page 2

- It would be useful to clarify the meaning of "incidence" as it varies between disciplines, or use another term

With incidence, we mean "the number of diseased plants in a population relative to the total number assessed". The manuscript has been edited to explain this terminology that is standard in Epidemiology, but maybe not in other fields. Nonetheless, we have revised the manuscript to clarify that the growth rate is the relevant part to determine the risk of establishment and not the absolute numbers, which, in addition, cannot be accurately forecasted in exponentially growing diseases as shown recently in (Castro et al., PNAS 117, 26190 (2020), Rosenkrantz et al., PNAS 119, e2109228119 (2022)).

- I don't understand the authors' points about "early infections" - can this be clarified please?

Thank you for pointing this out. After reading again the phrase, we realise it is not clear; we thus have changed it to: "*Because new infections late in the growing season are more likely to recover during winter than early infections, the*

phenology of the vector has a great influence on the dynamics of chronic infections and PD transmission [30, 42-44] ”

- "none of these works has explicitly included" should be "none of these works have explicitly included"

We have applied this change as suggested by the reviewer.

- I am not sure what the authors mean by "invasive criterion" in the context used here. Can this be clarified please?

This was not clear in the original manuscript. We have clarified this concept by replacing "invasive criterion" with:

"We follow an invasive criterion as defined by Jeger & Bragard [56] to include, as far as we can, key plant, pathogen, and vector parameters and their interactions for estimating the risk of establishment, persistence, and subsequent epidemic development."

- The wording "By reducing the global risk of PD" is misleading (since the true global risk of PD is unchanged). Something like "By estimating a lower global risk of PD" would be more correct.

This is a good point and we agree. It has been changed in the manuscript as suggested by the reviewer.

- "chronic infections" is not defined when it is first introduced. Could the authors define what they mean by this please?

Added for clarity. "Chronic infections, i.e., those that persist from one year to the next" in the revised manuscript.

Page 3

- The caption for Figure 1B does not explain what the lines represent (presumably they are means). Could this be stated please?

Thank you for spotting this. We have included "*Red solid line depicts the fitted exponential function for worldwide data and blue solid line for main vineyards zones*" in the new version of the manuscript.

Page 5

- It is a little unclear to me what "full presence of the vector" really means. Does it mean that vector densities would be expected to be maximal? Or that there are no climatic constraints on vector population sizes? I think that this should be explained as this is the estimate being used to scale the R0 value.

We agree this needs to be clarified. We now have included in the main text:

“ $v=1$ implies optimal climatic conditions for the vector with no constraints for the population size, while $v=0$ implies unsuitable climatic conditions and its absence”.

In addition, we have added to the Methods.

*“We used the positive relationship found between the climate suitability and abundance of *P. spumarius* adults [35] and assumed no climatic constraints on vector population sizes at the optimal climatic conditions ($v = 1$). Climatic suitability indices, $v(x)$, were used to compute a spatially-dependent basic reproduction number, $R_0(x)=R_0 \cdot v(x)$.”*

To justify this assumption, we show an analytical derivation of the linear dependence between R_0 and the vector population (i.e. the number of vectors) in the Supplementary Information (Sec. S2F and S4 in the SI). Then, assuming that climatic suitability (i.e. probability of the presence of the vector) is directly related to the number of vectors we obtain the linear scaling between R_0 and climatic suitability for vectors.

Page 7

- The paragraph starting "knowledge" starts in lower case. I assume that this is just a typo rather than words being missing.

This is certainly a typo, and has been amended as suggested by the reviewer.

- I think that the statement that there is "enough information on the principal vector" in Europe is debatable (this word could just be removed). Also, I suspect that this should be citing Figure S10 rather than S7.

The reviewer is right in that this statement is ambiguous. We have changed this to “*except for Europe where the relative distribution and abundance for the main vector *Philaenus spumarius* is well-documented*” (Figure S8)”.

- "indirect empirical evidence of this non-linear relationship" is better phrased as "indirect empirical evidence of a non-linear relationship"

We acknowledge the suggestion of the reviewer and have amended this phrase following his/her suggestion.

Page 8

- "our risk maps go further beyond by incorporating" is better phrased as "our risk maps go further by incorporating"

We acknowledge the reviewer for pointing out this type, which has been corrected in the revised manuscript.

- I think that the statement "a comparison of PD risk maps for Europe with different R_0 suggests for non-Mediterranean areas the need to stress more surveillance on the introduction of alien vectors rather than in the pathogen itself" requires more explanation.

We agree it needs further explanation. We have rephrased and added a little more information:

*"Remarkably, a comparison of PD risk maps for Europe with different R_0 suggests for non-Mediterranean areas the need to stress more surveillance on the introduction of alien vectors rather than in the pathogen itself. This is because, under the current scenario ($R_0 = 5$) with *P. spumarius* as the main vector, most of the non-Mediterranean vineyards thrive in non-risk zones, while the introduction of new insect vectors with greater transmission efficiency ($R_0 \geq 8$) could compensate the climatic layer and push the risk index above 0."*

Page 9

- "biweekly" is unclear as it can mean twice weekly or every other week. Could the authors clarify which of these is meant?

Changed for:

Every two weeks.

- "The mathematical relation bacterial population growth" should be "The mathematical relationship between bacterial population growth"

Thank you for the correction. Changed accordingly.

- Unless I have just missed it, I cannot see where the authors describe how CDD is captured in this Methods section. Can this be included please?

We acknowledge the comment of the referee and we agree with it. This was not well explained in the methods section. We have added a new subsection inside methods to explain it clearly.

"Disease recovery through winter curing.

To capture the accumulation nature of the chilling process and differences in climate zones, we determined the global average correlation between T_{min} and CDD using 6,487,200 points distributed throughout the planet. We found an exponential relation, $CDD \sim 230 \cdot \exp(-0.26 \cdot T_{min})$, where specifically, $CDD > 306$ correspond to $T_{min} < -1.1$ °C. To transform this exponential relationship to a probabilistic function analogous to $F(MGDD)$ ranging between 0 and 1, we considered the sigmoidal family of functions $f(x) = A/(B+x^C)$ with $A=9 \cdot 10^6$, $B=A$ and $C=3$ (Fig. 1C), fulfilling the limit $G(CDD=0)=1$, i.e. no winter curing when no cold accumulated, and a conservative 75% of the infected plants recovered at $T_{min} = -1.1$ °C instead of 100% to reflect uncertainties on the effect of winter curing.

Page 10

- The gamma in equation 1 is not defined until later in the text - can the authors define it here please?

We have corrected it in the new version of the manuscript, defining gamma, the inverse of the mean time for host death.

- I think that the authors should point out after "a spatial dependent $R_0(j)$ was incorporated" exactly how this was done (i.e. the product of R_0 and spatially-dependent climate suitability for vectors).

We acknowledge the comment of the referee; this point has been modified as suggested by the reviewer in the revised manuscript.

- I think that "at each time" would be better phrased as "at each timestep" to point out that a discrete time model was used here

We agree with the reviewer and, accordingly, have changed it to "at each time-step".

- "P. spumarius distribution [35] and" is missing an left parenthesis before the reference.

We have corrected this typo as suggested by the reviewer.

Page 12

- The site <http://pdrisk.ifisc.uib-csic.es/> does not appear to be accessible to the public. Can this be corrected before publication please?

We have checked that the web server was down, and have acted on it, so that the site is publically available again (i.e. without credentials).

Throughout the MS

- I also note that SI sections are sometimes referenced explicitly and sometimes as "SI Appendix". Could one or the other be used consistently please?

The reviewer is completely right, and this issue has been addressed so that the SI Appendix is now cited consistently as Supplementary Information.

SI

Page 7

- The sentence "and thereby the bacterial population after a given time t is related to the MGDD by Eq. (S9)" would be better placed directly after S9 than in its current position.

We acknowledge the comment of the referee. The reference should have been (S12) instead of (S9), and now the position is the correct one.

Page 8

- The term "continuous factor" is confusing to me, as factors are commonly considered as categorical. I think that "function" would make more sense. I also think that it would be useful to state that this function is bounded by 0 and 1. "Factors" are mentioned elsewhere in the text, and I think that these mentions should also be changed to "function".

We appreciate the reviewer's comment. Now "continuous factor" has been changed to "function".

- I think that the whole CDD paragraph needs to be rewritten/restructured (and the parameters explained) as I currently do not understand it.

We have rewritten/restructured the CDD paragraph. Part of the explanation has been also included in the material and methods section.

- The statement "With this choice, the 75% of the infected vineyard population get recovered in the winter curing threshold" should be rewritten as something like "With this choice, 75% of the infected vineyard population recover at the winter curing threshold"

We acknowledge the comment of the reviewer, and this point has been edited accordingly in the revised manuscript.

Page 9

- The authors should make clear whether the R_0 of 8 estimated for the USA is for ALS or PD.

We acknowledge the comment of the referee. We have revised this point in the new version of the manuscript and included it in the Result section "Model calibration and validation" as explained above.

I think that more explanation of the climate suitability for the vector estimates is needed - just a quick summary of how they were calculated and what exactly they mean.

We acknowledge the comment of the reviewer and this point has been further commented in the new version of the manuscript. We include the reply of the previous question.

We now have included in the main text:

" $v=1$ implies optimal climatic conditions for the vector with no constraints for the population size, while $v=0$ implies unsuitable climatic conditions and its absence".

In addition, we have added to the Methods.

“We used the positive relationship found between the climate suitability and abundance of *P. spumarius* adults [35] and assumed no climatic constraints on vector population sizes at the optimal climatic conditions ($v = 1$). Climatic suitability indices, $v(x)$, were used to compute a spatially-dependent basic reproduction number, $R_0(x) = R_0 \cdot v(x)$.”

To justify this assumption, we show an analytical derivation of the linear dependence between R_0 and the vector population (i.e. the number of vectors) in the Supplementary Information (Sec. S2F and S4 in the SI). Then, assuming that climatic suitability (i.e. probability of the presence of the vector) is directly related to the number of vectors we obtain the linear scaling between R_0 and climatic suitability for vectors.

- I do not understand why Equation S16 needs to be converted into a map in order to account for MGDD and CDD changing over time - could the authors clarify this please?

This is a good point indeed. The reason for writing the equation as an iterated map with discrete time equal to one year (in S17) is that the MGDD and CDD are accumulated for one year, or more precisely the MGDD from April to October and the CDD from November to March, and, thus, the functions $F(\text{MGDD})$ and $G(\text{CDD})$ are computed year to year. Thus, although S16 is defined for a continuous-time, S17 has been written as a map: it could not be defined for continuous time due to the way the MGDD and CDD are calculated.

- In the footnote at the bottom of the page, *Spumarius* is capitalised when it shouldn't be.

We acknowledge the reviewer for spotting this typo, that has been corrected.

Page 10

- The authors state that the upper limit of the Transition-risk zone is $r_{\{\max \text{ trans}\}} = 0.075$, but the lower limit of the Epidemic-risk zone is 0.1. Should this instead be $r_{\{\max \text{ trans}\}}?$

We acknowledge the reviewer for point out this, that is a typo. It should be $r_{\max \text{ trans}} = 0.075$, and has been corrected accordingly.

Page 11

- There are two question marks in the text.

This was a typo, that has been corrected.

Page 17

- There is a plus and a minus together in the equation of plot B. Although strictly speaking this is not incorrect, for consistency with plot A I think the plus can be removed. I also don't see the particular value of the intercept in either of these

plots - it would be more useful and informative to set it to a useful date (for example, 2019) rather than year 0.

We acknowledge the comment of the referee and this has been revised in the new version of the manuscript. However, we think that writing the equation as $y=m*(x-1981)-n$ can be confusing and we prefer to stick to the current form.

- It is very unclear to me what "the areas encompassed below the CDD ; 314 line" in Figure S6 means. Could the authors clarify please?

Thank for detecting this error; this was a technical issue with typesetting that has been corrected in the revised manuscript (to $CDD < 314$).

Page 18

- More information is needed in the legend of Figure S7 - such as the source of the data, the host, and so on. Can the authors add this in please?

We acknowledge the comment of the reviewer, which is completely right, because this is certainly not obvious. The sources of the data have been properly cited in the revised manuscript, now in the Results section "**Model calibration and validation**".

Pages 34-35

- Can the authors please sort the formatting of the titles of references 3, 4, 10 and 17 and the DOI and citation for reference 17?

We thank the referee for this comment. There was a technical problem that has now been solved.

While there are some improvements, unfortunately, it is difficult to say that the MS in its current state is much closer to recommendation for publication. There is an overwhelming feeling that although authors have done a lot to answer the critique from reviewers, the main points were only partially tackled. Also, second look at the MS reveals some other problems. I struggled to go through all these changes, and it took me more time than I expected and really wanted. Some of critique here could perhaps be coming from disciplinary differences and demands from the side of manuscript structure by the journal and conceptual framework for such type of a study. I still believe that I should not see so many surprises going through this study. I should be able to take a look and know how data is collected, model developed, applied and evaluated without any problems – which, if one takes a look in methods section for example, is not realistic. The free format of this study is frustrating, and the lack of structure needs to be addressed. I would suggest consultation with an experienced modeler from the field outside of the team and discussing these. Personally, I find the publications with methods at the end of the manuscript an unnecessary reminiscence of some past traditions which disturb the flow of manuscript such as this.

I can hardly see any improvement in presentation of the model. It is still split between several sections, of which briefest mention was in methods section. Why are there several sections in the results section devoted to model development? Furthermore, there are several segments presented in results section which were not mentioned in the methods section. The methods for validation were entitled to an entire sub-section in the results section as well as several references to disease data and no mentioning how this data was obtained. Was the literature search done? Is there public repository? Why selecting a specific part of the world for some things and other parts for other (China, Us, ...)?

I will echo once again my concerns which were not adequately addressed. Please beware that this is a simplified simulation model of the risk of disease spread and establishment in certain areas. Authors should be careful about suggesting the use of such modelling approach as a tactical IPM tool. Madden et al have published a good book on plant disease epidemiology and there are several other literature resources diving into differences and values of each tools.

Authors need to be transparent about the fact that the model is developed based on medium development data from a bacterial growth in growth medium in controlled environment. It must be noted that there are limitations with transferring developmental units from *in vitro* to *in vivo* conditions, especially having in mind, that these tests were done on growth media and only means were reported. This data is then modelled with a very unconvincing multilinear function. There is no organism responding to **the environmental conditions in a linear manner** and it is very **mathematically and biologically unconvincing** to see the first figure in the publication modelled in such manner.

Even if you insist on keeping the Why were regression lines drawn the way they were presented here? Why six lines? Why not draw a line through each point? (this is a rhetorical question)

The suggested (beta) function (or any other type of non-linear function such as polynomial) could be optimised to fit the data and used in the same manner as proposed (multi)linear function to facilitate more suitable biological representation of the pathogen biology. Furthermore, MGDD is then validated using weather data originating from **outside of the mesh tunnel**. Description of the weather station, sensors, distance from the site lack transparency and **must be explained in detail** in the main text.

There are only a few further recommendations as I gave up on detailed evaluation due to higher-level structural problems.

Pg1col2par3: Over praising the heat accumulation and GDD: Please inform potential reader about known problems with these approaches.

I would suggest changing illness to disease. This is term used in plant pathology. Term ill is more related to a more conscious state in humans or animals.

Methods

Inoculation tests: What does it mean: exposed to air temperature? This is nonsensical statement which might be originating from language barrier. Authors have failed to provide information about the weather station distance location and sensor description.

There are several sentences in the text where authors are commenting about success of their model in predicting the establishment of the disease or stating facts about regional distribution of grapevine production areas:

Emerging wine-producing areas in China are predominantly located in non-risk zones, whereas only some vineyards in the Henan province fall in transition risk zones (Fig. 3B).

All known areas where Xf is well-established in Europe (e.g. Apulia, Corsica, Balearic Islands, Region of Provence-Alpes C^ote d'Azur (French Riviera), Alicante) are in the 96th percentile of the tracked sites, validating the strength of our mechanistic, non-correlative PD model predictions (test in [57]). This is a study about possibility of establishment of PD.

Cold waves periodically occur that reach latitudes close to the Gulf, such as those that occurred in 1983, 1993, 1995, 2000, 2009 and 2013 (Movies at [57]), thus preventing PD expansion northward.

At no point in methods there was a description of the evaluation methods heavily mentioned in the results of the study. Such statements could be more appropriately placed in the discussion, if they were introduced and explained in to a reader.

I would strongly suggest English language check, but only prior to final publication. The MS is understandable but often not up to highest publication standard. An example

Semantics and some words: *Temperature **rules** key physiological processes in ectothermic organisms involved in PD and thus determines the ranges of thermal limitation in which they can thrive*

Supplementary materials: Data Analysis: AUDUP???

#####

Reviewer #2

This study attempts to predict how the risk of Pierce's Disease (PD) in grapevine varies spatially on a global level, with a particular focus on grapevine production areas. The novelty of this study compared to previous work is the use of a mechanistic model informed by the transmission dynamics of the pathogen rather than a purely correlative approach. This is a novel approach for this important pathogen, and may increase the robustness of the results and allow better investigation of important factors influencing pathogen distribution.

We thank the referee for this comment.

The study brings together a wide range of data - ranging from experimental data at the individual plant level up to satellite data at the global level - and with this, a range of analysis approaches. For this, the authors are to be commended. However, I found that clear explanation of the approaches used is sometimes lacking in both the main MS and in the SI. One major suggestion is therefore that the authors better describe their methods. However, this also means that there is a risk that I have misunderstood some aspects of the study, for which I apologise to the authors in advance.

We acknowledge the reviewer for this constructive remark. Actually, the model is an integral part of the Results of this work, and, so, following also the remarks of Referee #1, we now present the model in the Results section, before exposing the risk predictions. Further detailed aspects and technical discussions can be found in the Methods section and Supplementary Information. We hope that this order will allow a better understanding of the model and its predictions.

I agree with this change. I have still had a lot of difficulty understanding the approaches taken, which would have been lessened if the Methods had been moved to their conventional place in the MS, but I accept that other readers may be less interested in these details being upfront.

My main concerns about the study are as follows:

1. The authors argue that their experimental data provides valuable information on the temperature requirements for development of PD (which is captured in their MGDD parameter). However, as far as I can see, the authors are just monitoring the development of symptoms following inoculation with *X. fastidiosa*. I accept that the temperature needs to be appropriate for the pathogen to multiply and these symptoms to develop, but I have trouble seeing how the authors can argue that this development of symptoms directly reflects the influence of MGDD (that is, it is plausible that very similar patterns could potentially be observed with very different accumulations of MGDD). I appreciate that the author's approach is also supported by in vitro evidence of the influence of MGDD, and I suspect that the authors' approach is a reasonable one, but I feel that a little caution in the interpretation of MGDD as the cause of the disease progression (and a little discussion of how this uncertainty would impact upon the conclusions) would be useful, given the influence this could have on the model predictions.

This is indeed a very good point! Considering the temperature as the sole climatic influence that affects the development of Pierce's Disease, and neglecting other effects, like humidity and others is an approximation.

Although correct, this is not what I was saying. I have maybe misunderstood exactly how the authors have captured the relationship between MGDD and symptom expression. Originally, I thought that Figure 1C was showing how the authors would expect disease to develop (for example, in a single plant) as MGDD accumulated. However, I am now less sure. I would like the authors to explain more clearly exactly what Figure 1C shows. I would also like them to reference where exactly in the Supplementary Information the relevant information is. For example, I assume that the "chronic infections" are five symptomatic leaves. Is at some set time after inoculation? What exactly are the 15 MGDD levels shown? Do each of these contain different grapevine varieties?

Ideally, one would perform similar inoculation experiments like those described in the manuscript in different wine-producing regions in the world with different climate patterns to validate the predictions. It will be certainly interesting if such inoculation experiments would be carried out in the future. Given the impossibility of doing inoculations in different continents and climates, the closest approach is to verify how in the field the appearance of PD symptoms correlates with the accumulation of MGDD in other areas where PD is established. Thus, we have compared the predictions of the model in some spots of the Mediterranean basin with wine-producing locations in California and Oregon, finding that the model is quite predictive. Focusing on the MGDD in a single summer season, we observe a good correspondence between the MGDD in the inoculation experiments and PD symptom development in vineyards of Majorca in late July (cf. Fig 1a). This area is characterized by relatively hot and dry months in June and July, during which MGDD grows relatively fast until the value that yields PD with high probability. In contrast, the wine-producing areas of Napa and Sonoma valleys in California and also wine-producing areas in Oregon are characterized by a different Mediterranean climatic pattern. Namely, there is less heat accumulation at the beginning of the summer due to local fogs. Indeed, PD symptoms manifest in the late summer in Napa and Sonoma, and it is not observed in cool summers. In Southern Oregon the MGDD does not reach the threshold value for PD, and, in fact, PD is not observed (**Figure 2**). Although we agree that this is not a demonstration, we find that MGDD consistently predicts the appearance of symptoms of PD in these two areas. This can now be verified on our web page [57]. Finally, in 2000 the vector *Homolodisca vitripennis* was detected in Willamette Valley (Oregon), and this did not lead to a PD outbreak. Our model (with $R_0=8$) correctly predicts that the MGDD in this area does not lead to PD establishment.

Figure 2. Comparison of MGDD in Mallorca, northern California and Oregon. MGDD values above 1000 are approximately reached one month later in California than in Mallorca which shows a good correspondence with the lag of symptoms in the field. When included the probability of chronic infection (right) the differences widen. In Oregon, the climatic layer is not enough to produce PD.

We have also included in the Discussion the following sentence to reflect the uncertainties:

“The physiological basis of the plant-Xf interaction leading to symptoms development is poorly understood. Although the temperature is indeed a fundamental and dominant environmental factor, others such as drought, nutrient status or crop management may modulate symptom expression and hence add an error in the MGDD parameter not measured in this work. Nonetheless, we deem the error range would be smaller than the differences in MGDD among varieties (i.e., regional differences) and similar to the interannual MGDD oscillations in most locations”

I am not sure why the authors mention differences in MGDD among varieties. Can they clarify what the mean here?

2. I have a lot of trouble understanding how the authors are capturing CDD, as the paragraph explaining this in the SI jumps back and forth between different ideas. As far as I can see, it appears that the authors are arguing for the use of the average minimum temperature of the coldest month as a proxy for CDD, since both are correlated. However, I may have misunderstood this and feel that a better explanation is required here. It is also not explained how the function used for CDD is calculated, and upon which data this is based.

We recognize that the CDD part of the model describing the effect of winter curing was not as detailed and comprehensive as the MGDD in the submitted version of the manuscript.

Regarding CDD, the available information in the literature (Lieth et al.) is not enough to build a function capturing the probability of recovery. We had thus to

rely on the CDD measures, the standard degree-days accumulated below the base temperature of 6°C suggested in [6]. Additionally, we have checked whether this CDD metric is compatible with the main observation about winter curing of Purcell [40]: PD is not found in zones where the minimum average temperature of the coldest month is below -1.1°C. On the other hand, we have calibrated the CDD function in such a way that at the CDD threshold 75% of infected plants are cured, and that the combined effect of both the F(MGDD) and G(CDD) probability functions yields results that are compatible with the observed distribution of PD in the southeast of the United States.

Accordingly, we have expanded the explanation in the revised manuscript: “We then investigated the probability of disease recovery by exposure to cold temperatures. However, unlike the approach followed to relate MGDDs to $X_{f_{PD}}$ growth and symptom development, there are sparse experimental data to lean on. Following the work of Lieth et al. and an inverse analogy with the MGDDs, we assume that cold temperatures contribute to the decrease of the bacterial population in planta. Thus, the accumulation of cold degree-days (CDD) is expected to be a proper metric

What do the authors mean by “proper metric”?

to describe the probability of recovery of infected plants. To account for the effect of winter curing, we related the accumulation of cold degree-days (CDD with base temperature = 6 C, i.e., $CDD(t) = \sum_i 6 - T_i$ for $T \leq 6$ C) with the average minimum temperature of the coldest month, T_{min} , isolines in the United States proposed by Purcell (available in [41]) as reference zones where PD is rare (T_{min} between -1.1 C and 1.7 C), occasional (1.7- 4.5 C) and severe (> 4.5 C) (Fig. 1B, Methods). This allows to relate accumulated CDD to the probability of disease recovery) using a generalised sigmoidal function, G(CDD) (see blue curve in Fig. 1C and Methods).”

The authors make their approach fairly clear in their response to me, but appear to be determined to obfuscate it in the explanation in the MS itself. This is apparent throughout the MS, and to be honest makes reading it a little frustrating. I would much rather have an MS where the approaches were clearly explained and described.

3. The discussion of R_0 estimation is also a little unclear. As I see it, the authors use at least two different approaches to estimate R_0 (for ALS in Europe, PD in the USA, and potential PD in Europe), but these are minimally explained. Presumably the method based on the models (used for the European estimates for ALS and PD) involves the analysis of next generation matrices, but if so the method should be briefly explained.

We admit this part needed more explanations in the main text. The analytical derivation of R_0 can be consulted in the Supplementary information. Although R_0 could be calculated with next-generation matrices, we did it directly from the linear stability analysis of the fixed point, as it is easier for the SIR model.

All the R_0 values used throughout the manuscript are for PD;

What about Figure S6?

the only difference is how they were estimated. In the USA, given that there are available records of PD distribution, a ROC curve was computed to assess the model accuracy under different R_0 scenarios, calibrating the model. This approach could not be followed for Europe due to the lack of data. An alternative estimation taking as basis ALSD was performed. This is now explained in the Results in the model calibration and validation section:

“Model calibration and validation. To attempt a rough estimate of the R_0 parameter in the United States assuming a uniform spatial distribution of the vectors, we ran several model simulations validating the spatiotemporal distribution of PD from data collected from publications between 2001 and 2015. We found $R_0 = 8$ as the optimal parameter for maximising the area under a ROC curve (Supplementary Fig. S4). In general, our model returns an accuracy of more than 80%, except for 2006, due to data retrieved from a study in the Piedmont region in the USA at the limit of the transient-risk zone (Supplementary Fig. S6 and Table III). A different approach was followed to estimate R_0 for Europe given that PD is only present in Majorca and hence spatiotemporal data on the PD distribution is not available. First, we inferred the transmission rate, $\beta = 0.8$ years⁻¹,

As a very minor point, I find this way of presenting rates very confusing, especially in some of the uses below (e.g. 1/14 years⁻¹), as although mathematically sound it really doesn't read well. Could the authors not just use 0.8/year? I again refer back to my point above about the authors' repeated decisions to avoid clear explanations in favour of more complicated, obtuse, ones.

of the main European vector, *P. spumarius*, from the well-studied disease progress curve of the almond leaf scorch epidemic in Majorca and the estimated almond tree mortality rate, $\gamma \sim 1/14$ years⁻¹. Then, using the known mortality rate of PD-infected vines $\gamma \sim 1/5$ years⁻¹ and the inferred transmission rate, we could determine $R_0 = \beta/\gamma = 4$ for PD in Majorca. Finally, using data on the climate suitability of the vector in Majorca, $v = 0.8$, and inverting the relation $R_0(j) = R_0 \cdot v(j)$, we estimated a lower $R_0 = 4/0.8 = 5$ as a baseline scenario

Would this be better described as a maximal estimate than a baseline estimate?

for PD transmission in Europe (Supplementary Section S2 D). This figure is not intended to be an exact estimate of R_0 but rather an average reference in our model in agreement with the lesser abundance of vectors relative to the United States. Furthermore, since there is no information on the distribution of the potential vectors and no PD distribution data to calibrate, we also used a conservative $R_0=5$ scenario for the rest of the world.”

It may also be useful for the authors to point out to readers that they are not explicitly modelling vector infection, as the R_0 calculations (and likely the estimates themselves) would be different if so.

We acknowledge that our SIR approach does not explicitly account for the vector. However, if the decay-population time scales of hosts and vectors are very

different it can be reduced to a SIR model. This is explained now in S4 in the SI and recently shown in a publication [arXiv:2202.05598](https://arxiv.org/abs/2202.05598) by two of the authors for a quite general vector disease model with host-vector interactions. This is the case of PD in which the lifetime of the vector and the time span of the disease progress on the plant are very different, which gives ground to using the SIR model in our study.

I am happy with this response.

4. I also have some concerns about the assumption that the R_0 is either fixed or scales linearly with vector climate suitability. Here, the authors are capturing a lot of complex processes (including vector feeding behaviour and movement) in a very simple way. Although I do appreciate that such simplifications are needed in models such as this (and again, I do think that it is a justifiable assumption for the authors to make), I think that a brief mention/discussion of this assumption would be useful.

As pointed out by the referee, the linear scaling of R_0 with vector climate suitability needs to be clarified. In several compartmental models of vector-borne diseases, the basic reproduction number (R_0) scales linearly with the vector population, which justifies our assumption. In Section 4 of the Supplementary Information, we have included an analytical derivation of the linear dependence between R_0 and the vector population (i.e. the number of vectors). Then, assuming that climatic suitability (i.e. probability of presence) is directly related to the number of vectors we obtain the linear scaling between R_0 and climatic suitability for vectors.

I am happy with this response.

Some other minor points are listed below, indexed by page number:

We acknowledge the referee for his/her detailed reading of the manuscript and for pointing out these typos and other minor comments, that certainly improve the manuscript.

Unless otherwise indicated, I am happy with these responses.

Page 1

- "> 560 plants species" should be "> 560 plant species"

This typo has been corrected.

Page 2

- It would be useful to clarify the meaning of "incidence" as it varies between disciplines, or use another term

With incidence, we mean “the number of diseased plants in a population relative to the total number assessed”. The manuscript has been edited to explain this terminology that is standard in Epidemiology, but maybe not in other fields.

It isn't standard in Epidemiology as a whole. It is standard in botanical epidemiology, but incidence has a very different meaning in human and veterinary epidemiology (as the rate of new infections, in contrast to the proportion of infections which the authors mean here and which is defined as the prevalence in these other fields).

Nonetheless, we have revised the manuscript to clarify that the growth rate is the relevant part to determine the risk of establishment and not the absolute numbers, which, in addition, cannot be accurately forecasted in exponentially growing diseases as shown recently in (Castro et al., PNAS 117, 26190 (2020), Rosenkrantz et al., PNAS 119, e2109228119 (2022)).

- I don't understand the authors' points about "early infections" - can this be clarified please?

Thank you for pointing this out. After reading again the phrase, we realise it is not clear; we thus have changed it to: *“Because new infections late in the growing season are more likely to recover during winter than early infections, the phenology of the vector has a great influence on the dynamics of chronic infections and PD transmission [30, 42-44] ”*

- "none of these works has explicitly included" should be "none of these works have explicitly included"

We have applied this change as suggested by the reviewer.

- I am not sure what the authors mean by "invasive criterion" in the context used here. Can this be clarified please?

This was not clear in the original manuscript. We have clarified this concept by replacing “invasive criterion” with:

“We follow an invasive criterion as defined by Jeger & Bragard [56] to include, as far as we can, key plant, pathogen, and vector parameters and their interactions for estimating the risk of establishment, persistence, and subsequent epidemic development.”

- The wording "By reducing the global risk of PD" is misleading (since the true global risk of PD is unchanged). Something like "By estimating a lower global risk of PD" would be more correct.

This is a good point and we agree. It has been changed in the manuscript as suggested by the reviewer.

- "chronic infections" is not defined when it is first introduced. Could the authors define what they mean by this please?

Added for clarity. "Chronic infections, i.e., those that persist from one year to the next" in the revised manuscript.

Page 3

- The caption for Figure 1B does not explain what the lines represent (presumably they are means). Could this be stated please?

Thank you for spotting this. We have included "*Red solid line depicts the fitted exponential function for worldwide data and blue solid line for main vineyards zones*" in the new version of the manuscript.

The authors have incorrectly pluralised many words in the MS – "vineyards" being the example in this sentence (which should be "vineyard").

Page 5

- It is a little unclear to me what "full presence of the vector" really means. Does it mean that vector densities would be expected to be maximal? Or that there are no climatic constraints on vector population sizes? I think that this should be explained as this is the estimate being used to scale the R0 value.

We agree this needs to be clarified. We now have included in the main text:

" $v=1$ implies optimal climatic conditions for the vector with no constraints for the population size, while $v=0$ implies unsuitable climatic conditions and its absence".

There is a typo here but it isn't present in the main MS.

In addition, we have added to the Methods.

*"We used the positive relationship found between the climate suitability and abundance of *P. spumarius* adults [35] and assumed no climatic constraints on vector population sizes at the optimal climatic conditions ($v = 1$). Climatic suitability indices, $v(x)$, were used to compute a spatially-dependent basic reproduction number, $R0(x)=R0 \cdot v(x)$."*

To justify this assumption, we show an analytical derivation of the linear dependence between R0 and the vector population (i.e. the number of vectors) in the Supplementary Information (Sec. S2F and S4 in the SI). Then, assuming that climatic suitability (i.e. probability of the presence of the vector) is directly related to the number of vectors we obtain the linear scaling between R0 and climatic suitability for vectors.

Page 7

- The paragraph starting "knowledge" starts in lower case. I assume that this is just a typo rather than words being missing.

This is certainly a typo, and has been amended as suggested by the reviewer.

- I think that the statement that there is "enough information on the principal vector" in Europe is debatable (this word could just be removed). Also, I suspect that this should be citing Figure S10 rather than S7.

The reviewer is right in that this statement is ambiguous. We have changed this to "except for Europe where the relative distribution and abundance for the main vector *Philaenus spumarius* is well-documented" (Figure S8)".

- "indirect empirical evidence of this non-linear relationship" is better phrased as "indirect empirical evidence of a non-linear relationship"

We acknowledge the suggestion of the reviewer and have amended this phrase following his/her suggestion.

Page 8

- "our risk maps go further beyond by incorporating" is better phrased as "our risk maps go further by incorporating"

We acknowledge the reviewer for pointing out this type, which has been corrected in the revised manuscript.

- I think that the statement "a comparison of PD risk maps for Europe with different R_0 suggests for non-Mediterranean areas the need to stress more surveillance on the introduction of alien vectors rather than in the pathogen itself" requires more explanation.

We agree it needs further explanation. We have rephrased and added a little more information:

*"Remarkably, a comparison of PD risk maps for Europe with different R_0 suggests for non-Mediterranean areas the need to stress more surveillance on the introduction of alien vectors rather than in the pathogen itself. This is because, under the current scenario ($R_0 = 5$) with *P. spumarius* as the main vector, most of the non-Mediterranean vineyards thrive in non-risk zones, while the introduction of new insect vectors with greater transmission efficiency ($R_0 \geq 8$) could compensate the climatic layer and push the risk index above 0."*

Page 9

- "biweekly" is unclear as it can mean twice weekly or every other week. Could the authors clarify which of these is meant?

Changed for:

Every two weeks.

- "The mathematical relation bacterial population growth" should be "The mathematical relationship between bacterial population growth"

Thank you for the correction. Changed accordingly.

- Unless I have just missed it, I cannot see where the authors describe how CDD is captured in this Methods section. Can this be included please?

We acknowledge the comment of the referee and we agree with it. This was not well explained in the methods section. We have added a new subsection inside methods to explain it clearly.

“Disease recovery through winter curing.

To capture the accumulation nature of the chilling process and differences in climate zones, we determined the global average correlation between Tmin and CDD using 6,487,200 points distributed throughout the planet. We found an exponential relation, $CDD \sim 230 \cdot \exp(-0.26 \cdot Tmin)$, where specifically, $CDD > 306$ correspond to $Tmin < -1.1$ °C. To transform this exponential relationship to a probabilistic function analogous to $F(MGDD)$ ranging between 0 and 1, we considered the sigmoidal family of functions $f(x) = A / (B + x^C)$ with $A = 9 \cdot 10^6$, $B = A$ and $C = 3$ (Fig. 1C), fulfilling the limit $G(CDD=0) = 1$, i.e. no winter curing when no cold accumulated, and a conservative 75% of the infected plants recovered at $Tmin = -1.1$ °C instead of 100% to reflect uncertainties on the effect of winter curing.

Page 10

- The gamma in equation 1 is not defined until later in the text - can the authors define it here please?

We have corrected it in the new version of the manuscript, defining gamma, the inverse of the mean time for host death.

- I think that the authors should point out after "a spatial dependent $R_0(j)$ was incorporated" exactly how this was done (i.e. the product of R_0 and spatially-dependent climate suitability for vectors).

We acknowledge the comment of the referee; this point has been modified as suggested by the reviewer in the revised manuscript.

- I think that "at each time" would be better phrased as "at each timestep" to point out that a discrete time model was used here

We agree with the reviewer and, accordingly, have changed it to "at each time-step".

- "P. spumarius distribution [35] and" is missing an left parenthesis before the reference.

We have corrected this typo as suggested by the reviewer.

Page 12

- The site <http://pdrisk.ifisc.uib-csic.es/> does not appear to be accessible to the public. Can this be corrected before publication please?

We have checked that the web server was down, and have acted on it, so that the site is publically available again (i.e. without credentials).

Throughout the MS

- I also note that SI sections are sometimes referenced explicitly and sometimes as "SI Appendix". Could one or the other be used consistently please?

The reviewer is completely right, and this issue has been addressed so that the SI Appendix is now cited consistently as Supplementary Information.

SI

Page 7

- The sentence "and thereby the bacterial population after a given time t is related to the MGDD by Eq. (S9)" would be better placed directly after S9 than in its current position.

We acknowledge the comment of the referee. The reference should have been (S12) instead of (S9), and now the position is the correct one.

Page 8

- The term "continuous factor" is confusing to me, as factors are commonly considered as categorical. I think that "function" would make more sense. I also think that it would be useful to state that this function is bounded by 0 and 1. "Factors" are mentioned elsewhere in the text, and I think that these mentions should also be changed to "function".

We appreciate the reviewer's comment. Now "continuous factor" has been changed to "function".

- I think that the whole CDD paragraph needs to be rewritten/restructured (and the parameters explained) as I currently do not understand it.

We have rewritten/restructured the CDD paragraph. Part of the explanation has been also included in the material and methods section.

- The statement "With this choice, the 75% of the infected vineyard population get recovered in the winter curing threshold" should be rewritten as something like "With this choice, 75% of the infected vineyard population recover at the winter curing threshold"

We acknowledge the comment of the reviewer, and this point has been edited accordingly in the revised manuscript.

Page 9

- The authors should make clear whether the R_0 of 8 estimated for the USA is for ALS or PD.

We acknowledge the comment of the referee. We have revised this point in the new version of the manuscript and included it in the Result section “Model calibration and validation” as explained above.

I think that more explanation of the climate suitability for the vector estimates is needed - just a quick summary of how they were calculated and what exactly they mean.

We acknowledge the comment of the reviewer and this point has been further commented in the new version of the manuscript. We include the reply of the previous question.

We now have included in the main text:

“ $v=1$ implies optimal climatic conditions for the vector with no constraints for the population size, while $v=0$ implies unsuitable climatic conditions and its absence”.

In addition, we have added to the Methods.

“We used the positive relationship found between the climate suitability and abundance of *P. spumarius* adults [35] and assumed no climatic constraints on vector population sizes at the optimal climatic conditions ($v = 1$). Climatic suitability indices, $v(x)$, were used to compute a spatially-dependent basic reproduction number, $R_0(x)=R_0 \cdot v(x)$.”

To justify this assumption, we show an analytical derivation of the linear dependence between R_0 and the vector population (i.e. the number of vectors) in the Supplementary Information (Sec. S2F and S4 in the SI). Then, assuming that climatic suitability (i.e. probability of the presence of the vector) is directly related to the number of vectors we obtain the linear scaling between R_0 and climatic suitability for vectors.

- I do not understand why Equation S16 needs to be converted into a map in order to account for MGDD and CDD changing over time - could the authors clarify this please?

This is a good point indeed. The reason for writing the equation as an iterated map with discrete time equal to one year (in S17) is that the MGDD and CDD are accumulated for one year, or more precisely the MGDD from April to October and the CDD from November to March, and, thus, the functions $F(\text{MGDD})$ and $G(\text{CDD})$ are computed year to year. Thus, although S16 is defined for a continuous-time, S17 has been written as a map: it could not be defined for continuous time due to the way the MGDD and CDD are calculated.

- In the footnote at the bottom of the page, Spumarius is capitalised when it shouldn't be.

We acknowledge the reviewer for spotting this typo, that has been corrected.

Page 10

- The authors state that the upper limit of the Transition-risk zone is $r_{\text{max trans}} = 0.075$, but the lower limit of the Epidemic-risk zone is 0.1. Should this instead be $r_{\text{max trans}}$?

We acknowledge the reviewer for point out this, that is a typo. It should be $r_{\text{max trans}} = 0.075$, and has been corrected accordingly.

Page 11

- There are two question marks in the text.

This was a typo, that has been corrected.

Page 17

- There is a plus and a minus together in the equation of plot B. Although strictly speaking this is not incorrect, for consistency with plot A I think the plus can be removed. I also don't see the particular value of the intercept in either of these plots - it would be more useful and informative to set it to a useful date (for example, 2019) rather than year 0.

We acknowledge the comment of the referee and this has been revised in the new version of the manuscript. However, we think that writing the equation as $y=m*(x-1981)-n$ can be confusing and we prefer to stick to the current form.

- It is very unclear to me what "the areas encompassed below the CDD \leq 314 line" in Figure S6 means. Could the authors clarify please?

Thank for detecting this error; this was a technical issue with typesetting that has been corrected in the revised manuscript (to $CDD < 314$).

Page 18

- More information is needed in the legend of Figure S7 - such as the source of the data, the host, and so on. Can the authors add this in please?

We acknowledge the comment of the reviewer, which is completely right, because this is certainly not obvious. The sources of the data have been properly cited in the revised manuscript, now in the Results section "**Model calibration and validation**".

Pages 34-35

- Can the authors please sort the formatting of the titles of references 3, 4, 10 and 17 and the DOI and citation for reference 17?

We thank the referee for this comment. There was a technical problem that has now been solved.

Reply to Reviewers' Comments

We include the reviewers' comments in blue, point by point, and then develop our answer (in black)

While there are some improvements, unfortunately, it is difficult to say that the MS in its current state is much closer to recommendation for publication. There is an overwhelming feeling that although authors have done a lot to answer the critique from reviewers, the main points were only partially tackled. Also, second look at the MS reveals some other problems. I struggled to go through all these changes, and it took me more time than I expected and really wanted. Some of critique here could perhaps be coming for disciplinary differences and demands from the side of manuscript structure by the journal and conceptual framework for such type of a study. I still believe that I should not see so many surprises going through this study. I should be able to take a look and know how data is collected, model developed, applied and evaluated without any problems – which, if one takes a look in methods section for example, is not realistic. The free format of this study is frustrating, and the lack of structure needs to be addressed. I would suggest consultation with an experienced modeller from the field outside of the team and discussing these. Personally, I find the publications with methods at the end of the manuscript an unnecessary reminiscence of some past traditions which disturb the flow of manuscript such as this.

We appreciate the reviewer's comments and regret that the latest version and response letter have only partially solved the concerns raised. We have tried to delve into the reviewer's points of criticism, although in some passages we miss certain conciseness in the topics to be discussed. On the other hand, we agree that Methods would be better placed before Results, but this is a strict rule of the Journal and the editor has confirmed that it is not possible to modify the order.

In the new MS version, we have tried to clarify and improve the presentation of the field data and, subsequently, the development and parameterization of the epidemiological model. After careful thinking, we have chosen to place the disease model and all methodological material together in the Methods section ensuring that their sections match the Results section. We believe that the assumptions on which the model is based are now clearly established. We hope that with these changes and the explanations below, we can address the major concerns of the reviewer.

I can hardly see any improvement in presentation of the model. It is still split between several sections, of which briefest mention was in methods section. Why are there several sections in the results section devoted to model development?

The authors' consensus is that we would prefer to add a "Model Development" section before the Results, and then proceed with the description of some "standard methodologies" in the Methods. However, this is not an option, as mentioned above. In the former MS version, the disease model, risk modelling and model calibration and validation subsections were in the Results in order to facilitate the comprehension of the risk maps. Now, we have decided to include the disease model in the Methods, since

the description of the methods is much more detailed and the assumptions are better explained.

Furthermore, there are several segments presented in results section that were not mentioned in the methods section.

This fact is of major concern to us. As mentioned before, we have checked point by point the correspondence between the Methods and Results and moved some of the information from the Supplementary Information to the Methods section. We believe that in the new version, it is clearer and more detailed how data are collected for each of the topics addressed in the Results.

The methods for validation were entitled to an entire sub-section in the results section as well as several references to disease data and no mentioning how this data was obtained.

Now it has been changed with the restructuring. The methods for validation are in the Methods section, as well as the references on the distribution of PD in the USA. Now we further detail how these references were obtained:

“Model performance was calibrated with observed records of PD presence in California and the southeast of the US, where the disease is well established. PD distribution data were collected from publications from 2001 to 2020. Publications were filtered by selecting only records where the pathogen detection on symptomatic grapevines was confirmed by PCR or Elisa. The exact coordinates of the records were taken when available in the publication or approximated to locality or county level to build the Supplementary Data 1”

Was the literature search done?

All the information has been taken from the literature and cited in Supplementary Data 4.

Is there public repository?

No public repository has been created; collected information is included in Supplementary Data 4.

Why selecting a specific part of the world for some things and other parts for other (China, Us, ...)?

The validation of the model was based on the distribution of PD in the US because, as mentioned in the manuscript, PD is only present in the American continent (US grape wine), Majorca (Spain) and Taiwan. Similarly, we have only information about the vector in Europe. In the Methods, it is further emphasized that our study focuses on wine grapes and excludes table or dried grapes in the analyses. In the new

version, we provide information on the distribution of main grape-wine production regions and from where this information was collected.

“Distribution of wine-grape production areas. Risk maps focused solely on wine-grape regions excluding table and dried grapes producing areas. Data on the vineyard surface in Europe were obtained from the CORINE land-cover map [77]. Nomenclature of Territorial Units for Statistics (NUTS) was used as a geocoding for the subdivisions of European countries for statistical purposes. To visualize the locations of the main growing regions in the risk maps, we included dots representing the distribution of the main winegrowing regions collected from official statistics and maps from the countries”

I will echo once again my concerns which were not adequately addressed. Please beware that this is a simplified simulation model of the risk of disease spread and establishment in certain areas. Authors should be careful about suggesting the use of such modelling approach as a tactical IPM tool. Madden et al have published a good book on plant disease epidemiology and there are several other literature resources diving into differences and values of each tools.

Throughout the manuscript, we recall that the aim of the research is to provide approximate risk maps at a regional scale (continents) based on an epidemiological model. These maps contain a global picture of risk where, of course, some variation in risk on a local scale is expected. In the Discussion section, there is a full paragraph warning about the limitations of our model at the local scale due to differences in varieties planted, local vectors, crop management, irrigation, etc. We are also aware that some insect vectors could locally increase the disease incidence growth rate. All these factors contribute naturally to the uncertainty of the model and this point is discussed in the manuscript. The European case, in which there is less uncertainty about the potential vector involved and its distribution, is dealt with more in-depth, so the reader could understand the model’s strengths and flaws.

Integral Pest Management is out of the scope of the manuscript. We have modified the abstract to avoid potential misunderstandings in this issue.

Although it can be said that it is a simplified simulation model, we try to capture the limiting factors of PD development from a mechanistic approach supported by important experimental data. For example, the extent and robustness of the experimental data are unusual for the construction of risk maps –three-year inoculation trials and disease progress monitoring on 36 varieties, 57 rootstock/variety combinations and a total of 4430 measurements. Second, the simulations take into account an invasion criterion implicit in the epidemiological model. The model is composed of two layers, the transmission layer and the climatic layer. The effect of inter-annual climate variation and non-linear response to temperature is fully addressed. In addition, we maintain the temporal resolution (hourly mean temperature) for the metrics (MGDD, GDD, and T_{min} of the coldest month) from the epidemiologic model to the spatial risk maps. These spatial risk maps not only identify the potential for an outbreak but also the relative magnitude and rate of change of the infected population. As far as we know, no other risk maps for other plant diseases have taken this approach.

Moreover, we include three R_0 scenarios to address the uncertainties surrounding the vector. We show in detail the case of Europe, because the main insect vector *Philaenus spumarius* plays a central role in the transmission of the Xf-related diseases established in the continent. The European case illustrates how we can obtain more precise risk maps if the information of the vector is available. It also alerts of risk overestimation when the transmission layer is excluded from the model. In conclusion, though our risk maps show sharp risk gradients with respect to other SDM models or ecological niche models, it is at the same time quite conservative.

Authors need to be transparent about the fact that the model is developed based on medium development data from a bacterial growth in growth medium in controlled environment. It must be noted that there are limitations with transferring developmental units from *in vitro* to *in vivo* conditions, especially having in mind, that these tests were done on growth media and only means were reported.

We further remark in the manuscript that Xf's specific growth curve derives from a study carried out *in vitro* under non-limiting conditions and we also consider this limitation in the Discussion section. Although Feil & Purcell's (2001) publication only provides mean data, it is still a reference study in the "Xylella community", widely accepted and recommended in EPPO laboratory protocols, where the minimum, optimal and maximum temperatures for Xf growth are well defined.

We can understand the reviewer's concerns about the limitations of transferring developmental units from *in vitro* to *in vivo* conditions, as there has been reports of some differences in fungi and oomycetes (Chaloner *et al.* 2020). However, we would like to highlight that specific growth rate measures a quite different process in fungi than in bacteria: colony expansion in filamentous fungi and oomycetes versus single cell multiplication in bacteria. More importantly, disease development involves increases in bacterial load and movement of XfPD through xylem vessels (dead tissue), while fungal-caused disease development primarily involves spore germination, the penetration of plant barriers in the leaves, the growth of colonies in different plant tissues and direct interaction with a plant defense system. Consequently, disease development has quite a different significance in fungi than it does for Xf_{PD}/Pierce disease.

"This Modified Growing Degree Day (MGDD) profile enables us to measure the thermal integral from hourly average temperatures, improving the prediction scale of the biological process (Bütikofer *et al.* 2020). MGDD also provides an excellent metric to link Xf_{PD} growth in culture with PD development as, once the pathogen is injected into the healthy vine, symptoms progression mainly depends upon the bacterial load (i.e., multiplication) and the movement through the xylem vessel network, which are fundamentally temperature-dependent processes (Fry, 1990, Feil & Purcell 2001)"

This data is then modelled with a very unconvincing multilinear function. There is no organism responding to **the environmental conditions in a linear manner** and it is

very **mathematically and biologically unconvincing** to see the first figure in the publication modelled in such manner.

Even if you insist on keeping the Why were regression lines drawn the way they were presented here? Why six lines? Why not draw a line through each point? (this is a rhetorical question)

The suggested (beta) function (or any other type of non-linear function such as polynomial) could be optimised to fit the data and used in the same manner as proposed (multi)linear function to facilitate more suitable biological representation of the pathogen biology.

Certainly, the response of this bacteria is not linear, it follows an Arrhenius (exponential) behavior. Fig. 1A shows a linear approximation valid in a limited range of temperatures. In fact, in Eq. S3 we showed how this approximation can be performed. To avoid confusion, the new version of Fig 1A contains as an inset the Feil and Purcell's data in the original representation. This transformation is further explained in the new version of the MS. The fits to the response were inspired by the four lines introduced in Fig. 3 of Feil and Purcell's paper. The objective is to reproduce as close as possible the original empirical data.

Fig. R1. Differences between the Arrhenius based fit and the beta function representing X_f 's specific growth rate *in vitro*.

We thank the reviewer for his/her suggestion of using a beta function to fit the empirical points. We are aware that this type of functions has been successful used in fitting temperature dependence of in growth rates in other organisms. However, in this case the fit is not as good as can be seen in Fig. R1.

Fig R2. Risk maps obtained with the Arrhenius-based (top) and with the beta (bottom) functions. The model does not contain the vector layer

We have checked whether using a beta function produces changes in the risk indexes in the model and in the risk maps with respect to the Arrhenius-based approach. Since both curves produced different growth rates, one could expect some changes in the risk predictions. Still, to be able to fairly compare we need first to calibrate the model using the probability of developing chronic infections, as in Fig. 1C, with the values of MGDD obtained with the beta function. After doing so, some differences occur in the risk in specific areas, mainly those of transition between risk and non-risk zones (Fig. R2 and Fig. R3). The changes are not important at the global scale. The results obtained with the beta function tends to underestimate the risk in comparison to the Arrhenius-based function (Fig. R3). Properly approximating the empirical growth rates is important and, therefore, we will continue to use the Arrhenius-based function as it provides the best fit to the experimental growth data in the manuscript. Our model is general enough to admit other functions or fits would better growth data become available.

Fig R3. Risk maps differences in the risk indexes between the Arrhenius-based and the beta function.

Furthermore, MGDD is then validated using weather data originating from outside of the mesh tunnel. Description of the weather station, sensors, distance from the site lack transparency and must be explained in detail in the main text.

In the method section of the manuscript, we provide the requested information. We must clarify that the MGDDs are not validated in the sense mentioned by the reviewer. As explained above, MGDD is an ad hoc thermal metric for X_{fPD} , which is calculated using recorded hourly mean temperatures. The $F(MGDD)$ is calibrated with the inoculation data and from this function, the annual MGDD for each grid cell is calculated and then it is validated with the known distribution of PD in the US (see Table 1). As expected, $F(MGDD) > 90\%$ captures all PD records in tropical areas (Taiwan, Costa Rica, etc.).

There are only a few further recommendations as I gave up on detailed evaluation due to higher-level structural problems.

Pg1col2par3: Over praising the heat accumulation and GDD: Please inform potential reader about known problems with these approaches.

In the context of this paragraph and referring exclusively to Pierce's disease, this statement is not over-praising heat accumulation and GDD (see replies in other points). We do agree, however, that there are problems with using GDD when estimated with coarse climatic data (daily Tmax and Tmin) or used for biological processes in which other climatic data (leaf wetness for infection) are important. We

now inform of some of these problems reviewed by Bütikofer et al. (2020) in the method section and, how we try to avoid this, for example using GDHourly (though expressed as GDD) rather than GDD both for the “validation” (weather station with sensors at 2 m from the ground) and predictions (ERA-5 land hourly temperature surface temperature 2 meters above the ground).

I would suggest changing illness to disease. This is term used in plant pathology. Term ill is more related to a more conscious state in humans or animals.

Good point, now changed.

Methods

Inoculation tests: What does it mean: exposed to air temperature? This is nonsensical statement which might be originating from language barrier. Authors have failed to provide information about the weather station distance location and sensor description.

Thank you! We mean environmental temperature; it was a bad translation from Spanish. The automatic weather station registers among other variables, mean hourly temperatures with the sensor around two meters above the ground and located five meters from the entrance of the insect-proof tunnel. The ground surface below the station is a bare soil. Data used to build the risk maps, taken from the ERA5-Land Database, also correspond to temperatures two meters above ground.

There are several sentences in the text where authors are commenting about success of their model in predicting the establishment of the disease or stating facts about regional distribution of grapevine production areas

Now changed

“We found that emerging wine-producing areas in China are predominantly located in non-risk zones, whereas only some vineyards in the Henan and Yunnan provinces fall in transition and moderate-high risk zones (Fig. 3B and Supplementary Data 3).”

All known areas where Xf is well-established in Europe (e.g. Apulia, Corsica, Balearic Islands, Region of Provence-Alpes C^ote d’Azur (French Riviera), Alicante) are in the 96th percentile of the tracked sites, validating the strength of our mechanistic, non-correlative PD model predictions (test in [57]). This is a study about possibility of establishment of PD.

The reviewer is right. This is a study on the probability of establishment and epidemics of PD with respect to a specific pathogen XfPD. The sentence must be put in the correct context. Other Xf subspecies and clonal lineages show similar in vitro growth rates and cardinal temperatures, so similar results might be expected if the MGDD

approach is used in other crops. We have deleted this sentence from the manuscript as this information can be derived from Dataset 4.

Cold waves periodically occur that reach latitudes close to the Gulf, such as those that occurred in 1983, 1993, 1995, 2000, 2009 and 2013 (Movies at [57]), thus preventing PD expansion northward.

The movies of the annual GDD (1981-2019) are part of the results and clearly show how cold waves reach lower latitudes in the USA during those years. We indicate this to emphasize that inter-annual variation of CDD plays a significant role in preventing PD expansion.

At no point in methods there was a description of the evaluation methods heavily mentioned in the results of the study. Such statements could be more appropriately placed in the discussion, if they were introduced and explained in to a reader.

In the Method section, we now explicitly remark how the annual MGDD, CDD and average T_{\min} of the coldest month were globally computed for the 1981-2019 period. The MGDD, CDD and T_{\min} annual maps and movies are part of the Results and given on the webpage.

I would strongly suggest English language check, but only prior to final publication. The MS is understandable but often not up to highest publication standard. An example

Semantics and some words: Temperature rules key physiological processes in ectothermic organisms involved in PD and thus determine ranges of thermal limitation in which they can thrive

We have passed the manuscript on to some native speakers to minimize grammatical and idiomatic errors

Supplementary materials: Data Analysis: AUDUP???

Sorry for this mistake, now corrected: area under the disease progress curve (AUDPC)

References

Chaloner, T.M., Gurr, S.J. & Bebber, D.P. Geometry and evolution of the ecological niche in plant-associated microbes. *Nat Commun* 11, 2955 (2020). <https://doi.org/10.1038/s41467-020-16778-5>.

Bütikofer, L, Anderson, K, Bebber, DP, Bennie, JJ, Early, RI, Maclean, IMD. The problem of scale in predicting biological responses to climate. *Glob. Change Biol.* 2020; 26: 6657– 6666. <https://doi.org/10.1111/gcb.15358>.

Reviewer #2

I have still had a lot of difficulty understanding the approaches taken, which would have been lessened if the Methods had been moved to their conventional place in the MS, but I accept that other readers may be less interested in these details being upfront.

We agree that the approach would be clearer if the Methods section was moved upfront. Unfortunately, this is a strict rule of the journal and the editor has asked us to not invert the order of the Method and Results sections. Anyway, we provide in the new version of the Methods a comprehensive explanation of how the model is built, including the assumptions taken. We think that this will make the MS easier to understand.

I have maybe misunderstood exactly how the authors have captured the relationship between MGDD and symptom expression. Originally, I thought that Figure 1C was showing how the authors would expect disease to develop (for example, in a single plant) as MGDD accumulated. However, I am now less sure. I would like the authors to explain more clearly exactly what Figure 1C shows. I would also like them to reference where exactly in the Supplementary Information the relevant information is. For example, I assume that the “chronic infections” are five symptomatic leaves. Is at some set time after inoculation? What exactly are the 15 MGDD levels shown? Do each of these contain different grapevine varieties?

We now have revised this result section, the captions of the figure 1 and the methods to clarify the relationship between MGDD and symptoms expression. In the new version, it is clearly indicated that the number of symptomatic leaves was noted at the 8th, 10th, 12th, 14th and 16th week after the day of inoculation and that this was done in three independent trials performed in 2018, 2019 and 2020. This is why there are 15 fixed MGDDs (5 measure periods x 3 years) and what varies in Fig. 1C is the percentage of inoculated plants that reached the threshold of five or more symptomatic at the amount of MGDD for that period and year. The accumulated MGDD, for example, at the 12th week after inoculation in the 2018 trial was 678 (this can be consulted in Dataset S1). This figure was obtained by applying the $f(t)$ function to the mean hourly temperature data registered with the weather station located five meters outside the insect-proof tunnel. We also try to bring some light to the assumption of chronic infection.

I am not sure why the authors mention differences in MGDD among varieties. Can they clarify what the mean here?

Thank you for spotting this. To clarify this, we have modified the text as:

“Nonetheless, we deem the error range would be smaller than the differences in accumulated MGDDs needed to reach the same disease level among varieties (i.e., regional differences) and similar to the interannual MGDD oscillations in most locations”

We think this is now clear after the above response.

What do the authors mean by “proper metric”?

Indeed, it is not a fortunate expression. We have deleted it.

The authors make their approach fairly clear in their response to me, but appear to be determined to obfuscate it in the explanation in the MS itself. This is apparent throughout the MS, and to be honest makes reading it a little frustrating. I would much rather have an MS where the approaches were clearly explained and described.

We apologize for that; we think our explanation is now clearer in the new Methods section.

What about Figure S6?

Ok, we meant in the main MS. The R_0 in Fig S6 is for ALS, but we do not use this number as input of our model in any case. We use always the R_0 's for PD.

As a very minor point, I find this way of presenting rates very confusing, especially in some of the uses below (e.g. 1/14 years⁻¹), as although mathematically sound it really doesn't read well. Could the authors not just use 0.8/year? I again refer back to my point above about the authors' repeated decisions to avoid clear explanations in favour of more complicated, obtuse, ones.

Changed for more standard 0.07 y⁻¹; 0.8 y⁻¹

Would this be better described as a maximal estimate than a baseline estimate?

Completely agree with that! Now changed in the manuscript.

It isn't standard in Epidemiology as a whole. It is standard in botanical epidemiology, but incidence has a very different meaning in human and veterinary epidemiology (as the rate of new infections, in contrast to the proportion of infections which the authors mean here and which is defined as the prevalence in these other fields).

Thank you it is true and a good point.

The authors have incorrectly pluralised many words in the MS – “vineyards” being he example in this sentence (which should be “vineyard”).

We have passed the manuscript on to some native speakers to minimize grammatical and idiomatic errors.